SOFTWARE

# BharatSim: An agent-based modelling framework for India

**Philip Cherian**[1], **Jayanta Kshirsagar**[2], **Bhavesh Neekhra**[3], **Gaurav Deshkar**[2], **Harshal Hayatnagarkar**[2], **Kshitij Kapoor**[3], **Chandrakant Kaski**[2], **Ganesh Kathar**[2], **Swapnil Khandekar**[2], **Saurabh Mookherjee**[2], **Praveen Ninawe**[2], **Riz Fernando Noronha**[1], **Pranjal Ranka**[2], **Vaibhhav Sinha**[1,4], **Tina Vinod**[2], **Chhaya Yadav**[2], **Debayan Gupta**[3], **Gautam I. Menon**[1,5,6,7]*

**1** Department of Physics, Ashoka University, Sonepat, Haryana, India, **2** Engineering for Research (e4r), Thoughtworks Technologies, Pune, Maharashtra, India, **3** Department of Computer Science, Ashoka University, Sonepat, Haryana, India, **4** Simons Centre for the Study of Living Machines, National Centre for Biological Sciences, Tata Institute of Fundamental Research, Bangalore, Karnataka, India, **5** Department of Biology, Trivedi School of Biological Sciences, Ashoka University, Sonepat, Haryana, India, **6** The Institute of Mathematical Sciences, CIT Campus, Taramani, Chennai, India, **7** Homi Bhabha National Institute, Anushaktinagar, Mumbai, Maharashtra, India

* gautam.menon@ashoka.edu.in

**Data Availability Statement:** The code that generates the synthetic populations used in this paper is available on the BharatSim GitHub page (www.github.com/bharatsim). In addition, the

## Abstract

BharatSim is an open-source agent-based modelling framework for the Indian population. It can simulate populations at multiple scales, from small communities to states. BharatSim uses a synthetic population created by applying statistical methods and machine learning algorithms to survey data from multiple sources, including the Census of India, the India Human Development Survey, the National Sample Survey, and the Gridded Population of the World. This synthetic population defines individual agents with multiple attributes, among them age, gender, home and work locations, pre-existing health conditions, and socio-economic and employment status. BharatSim's domain-specific language provides a framework for the simulation of diverse models. Its computational core, coded in *Scala*, supports simulations of a large number of individual agents, up to 50 million. Here, we describe the design and implementation of BharatSim, using it to address three questions motivated by the COVID-19 pandemic in India: (i) When can schools be safely reopened given specified levels of hybrid immunity?, (ii) How do new variants alter disease dynamics in the background of prior infections and vaccinations? and (iii) How can the effects of varied non-pharmaceutical interventions (NPIs) be quantified for a model Indian city? Through its India-specific synthetic population, BharatSim allows disease modellers to address questions unique to this country. It should also find use in the computational social sciences, potentially providing new insights into emergent patterns in social behaviour.

## 1 Introduction

Mathematical and computational approaches to the dynamics of an infectious disease have a long history. Around a hundred years ago, Kermack and McKendrick [1], expanding on the

synthetic populations used in our simulations can be found on the BharatSim website (bharatsim. ashoka.edu.in).

**Funding:** GIM was supported by the Bill and Melinda Gates Foundation under Grant No: R/BMG/ PHY/GMN/20. PC and VS received support from the World Health Organization under Grant No: APW202706833. PC, VS, BN, KK, and RN received salary support from the MPhasis F1 Foundation. The funders had no role in the study design, data collection and analysis, decision to publish, or preparation of the manuscript.

**Competing interests:** The authors have declared that no competing interests exist.

ideas of Bernoulli, Hamer, and Ross [2–4], used a set of coupled non-linear ordinary differential equations to describe how the numbers of susceptible (*S*), infectious (*I*) and recovered (*R*) individuals in a population changed over time. Their model for the dynamics of an infectious disease is referred to as the *SIR* model. In this model, individuals are assigned to "compartments" labelled *S*, *I* and *R* [5]. Variants of the *SIR* model that incorporate additional compartments, such as Exposed (*E*) or Vaccinated (*V*), have been customized to different infectious diseases, as well as to varied interventions [6].

Compartmental models are valuable tools for the description of infectious disease dynamics. They can be formulated in terms of coupled non-linear ordinary differential equations (ODEs) and solved numerically with relative ease. However, in their most-used formulation, they are mean-field (also called "well-mixed") models. By construction, mean-field models are deterministic, excluding information contained in real-world networks of physical interactions and in spatial heterogeneities [6]. Social factors, such as family structures, socioeconomic status, and contact networks, are difficult to incorporate into such models. They are particularly ill-equipped to describe how population-level shifts emerge from subtle modifications in individual behaviour. An inability to easily describe the many independent sources of stochasticity that are relevant to disease dynamics is another limitation of these models.

Achieving a more comprehensive understanding of disease spread mandates modelling approaches that can go beyond the assumption of a well-mixed population [7–12]. At the most granular level, one could describe each separate individual in a population. Agent- and network-based models implement such granularity, with individuals in these models capable of modulating their behaviour in response to how they perceive their environment [13–16], including their processing of information obtained through their contact networks [17–19]. Such models can also incorporate geo-spatial data [20].

This increased flexibility comes at a cost. Both network- and agent-based models must use many more assumptions than compartmental models require. The parameters that enter these models are imperfectly known and their outputs are often sensitive to the precise values chosen. Studying such models also requires significant amounts of numerical computation.

On the positive side, however, more realistic investigations of disease dynamics in specific populations become possible [21]. The impacts of targeted interventions, such as lockdowns, school closures, vaccination drives, and restrictions on public transport, can be assessed more realistically than is possible with ODE-based approaches [22, 23]. An exponential expansion in computational power over the past several decades has also made simulating large-scale individual-level models progressively easier.

A number of agent-based models have been used to study disease dynamics, including of measles, Ebola, typhoid, and influenza [24–32]. Several have been developed and used for COVID-19 [33–36]. However, relatively few such models have been customized for India [37, 38]. One, the IISc-TIFR city-scale agent-based simulator [39], describes COVID-19 spread in the major Indian cities of Bengaluru and Mumbai. This simulator has been used to evaluate strategies for the reopening of public transport in the background of an epidemic. It has also been used to study lockdowns and related interventions. Another example of a Indian city-scale simulator is *EpiRust* [40, 41], which has been used to simulate the cities of Mumbai and Pune.

Here, we describe BharatSim, an open-source, ultra-large-scale, agent-based framework for the simulation of infectious diseases. This simulation framework provides a synthetic population that is customized to India, although it is easily generalizable to other countries and contexts. BharatSim allows users to easily adapt their models to different infectious diseases without having to modify a pre-existing code base.

Typically, modellers run such simulations multiple times to obtain statistically meaningful results. Consequently, one should be able to efficiently scale up a model without compromising on performance. This is often challenging because memory requirements and computational time in agent-based models generically increase non-linearly with the number of agents and the detail with which they are described. The BharatSim simulation engine is designed keeping such requirements in mind. Finally, BharatSim's formulation as a simulation modelling framework, as opposed to an application, insulates the user from the details of its implementation, while providing them flexibility in the construction of models.

A typical user of BharatSim will specify a model, initialize it with input data, run the simulation, and then analyse its results, often visually. BharatSim's design incorporates three components: a synthetic population designed for India, a simulation engine, and a visualization engine. A user might use the simulation engine to construct and simulate a model of disease spread on any given synthetic population. They could then visualize the output from the simulation using the visualization engine. However, these components can also be used independently.

In this paper, we describe the generation of the synthetic population, and the design of the simulation and visualization engines that constitute BharatSim. To validate our results, we compare statistical measures applied to the synthetic population to real-world survey data. We describe the different components of the simulation engine and show how it can be used by a typical user to create a customized model. We detail the features of the visualization engine, including the variety of spatio-temporal data that can be imported and visualized. We then explore how the impact of policy interventions can be assessed by simulating multiple scenarios, including counter-factual ones.

We apply BharatSim to three questions in the context of the COVID-19 pandemic: (i) Under what conditions can schools be safely reopened given specified levels of hybrid immunity?, (ii) How do new SARS-CoV-2 variants that evade prior immunity alter disease dynamics? and (iii) How do we quantify the effects of varied NPIs on the spread of the disease in a model Indian city? For this, we use disease states described in a specific compartmental model for COVID-19 called INDSCI-SIM, a model has been benchmarked using data for the first wave of the disease in India [42].

We show that at reported levels of seropositivity in India, and given the relative mildness of COVID-19 infections in the younger population, schools across India could have been reopened as early as August of 2021 with only a marginal increase in the number of cases. Further, our models show that the relative mildness of the Omicron wave was a consequence of hybrid immunity. Finally, we describe the impacts of targeted NPIs such as lockdowns and vaccination drives. We show that, when judiciously applied, such interventions contribute significantly and in a synergistic manner, to slowing the progress of such a disease.

Finally, to demonstrate the flexibility of BharatSim to model the spread of a qualitatively different disease, we consider the case of mpox, a largely sexually-transmitted disease. Studies [43, 44] have indicated the possibility that mpox may be transmitted through alternative routes that do not involve sexual contact, for example, contact with bodily fluids. (This study is described in detail in S9 Appendix.) Here, accounting for the network structure of the MSM population is essential, as is capturing the sub-networks of interactions between the MSM and non-MSM populations.

## 2 Design and implementation

BharatSim is designed to be usable on a range of hardware, from personal laptops to High Performance Computing (HPC) clusters. Given that one application of BharatSim would be the

simulation of an average Indian state ($\sim$ 50 million agents), it is designed to handle large population sizes without significant overhead or degradation in speed (see S1 Appendix). Its design allows modellers from a range of backgrounds, even those lacking significant programming experience, to be able to easily run existing models, tweak them and extend them in new ways. BharatSim is accessible at the site: bharatsim.ashoka.edu.in.

## 2.1 Synthetic population

BharatSim's synthetic population models those details of the actual population that are relevant to the interactions between agents. We use a variety of data sources to generate a population of individuals and households with demographic attributes that are statistically similar to survey data, using statistical methods and machine learning algorithms. These can be used to generate data at various scales, ranging from small communities to entire states [45].

**2.1.1 Datasets and data preparation.** The primary sources of data for our algorithms include the Census of India, the India Human Development Survey (IHDS), the National Sample Survey (NSS), and the Gridded Population of the World (GPW). We use sample survey data (microdata) from India Human Development Survey-II [46], marginals from the 2011 census [47], data for employment from NSS [48], and population density from GPW grid population density dataset [49]. A hybrid process is used to scale up these limited datasets to the size of a population. The data is initially curated to remove obvious inconsistencies. Then, the customized hybrid of statistical and machine learning techniques detailed below is used to generate new data on the required scale. Table 1 lists the different datasets and the role they play in the generation of the synthetic population.

The IHDS-II (2011–12) [46] is a nationally representative, multi-topic survey of 42,152 households in 1,503 villages and 971 urban neighbourhoods across India. We use microdata from two datasets from this survey, *Individual* and *Household*, in the machine-learning model described below. The 2011 census was performed across 28 states and 8 union territories, covering 640 districts, 5,924 sub-districts, 7,935 towns and more than 600,000 villages. We use marginal data from this census as input to the Iterative Proportional Updating (IPU) method described below, since the census microdata is not publicly available. The NSS collects data through nationwide sample household surveys on various socio-economic subjects. We use employment data from this survey. GPW provides spatial population maps for all countries and their sub-divisions—we use population density data for India to place the generated synthetic population in a geographical context. The next section describes the use of these different datasets in our hybrid model to generate a synthetic population for a district of India.

**2.1.2 Population generation.** Our objective is to generate an accurate synthetic population for India with joint distributions of relevant variables that statistically match those of the actual population. The synthetic population should capture relevant attributes of individuals in the real population, such as the distributions of age, height, weight, and sex. The synthetic population must also incorporate social network structures appropriate to households and workplaces. We might also require the distribution of family sizes, the joint distribution of age

**Table 1. Source datasets for generating the synthetic population.** The different data sources used are specified, along with the role they play in the generation of the synthetic population.

| Source | Description | Use in model |
|---|---|---|
| IHDS [46] | State-wise microdata | Generating attributes for individuals and families |
| Census [47] | District-wise marginal data | Generating attributes for individuals and families |
| NSS [48] | District-wise data for employment | Estimating homebound population |
| GPW [49] | Population density data | Assigning geo-locations to households and other locations |

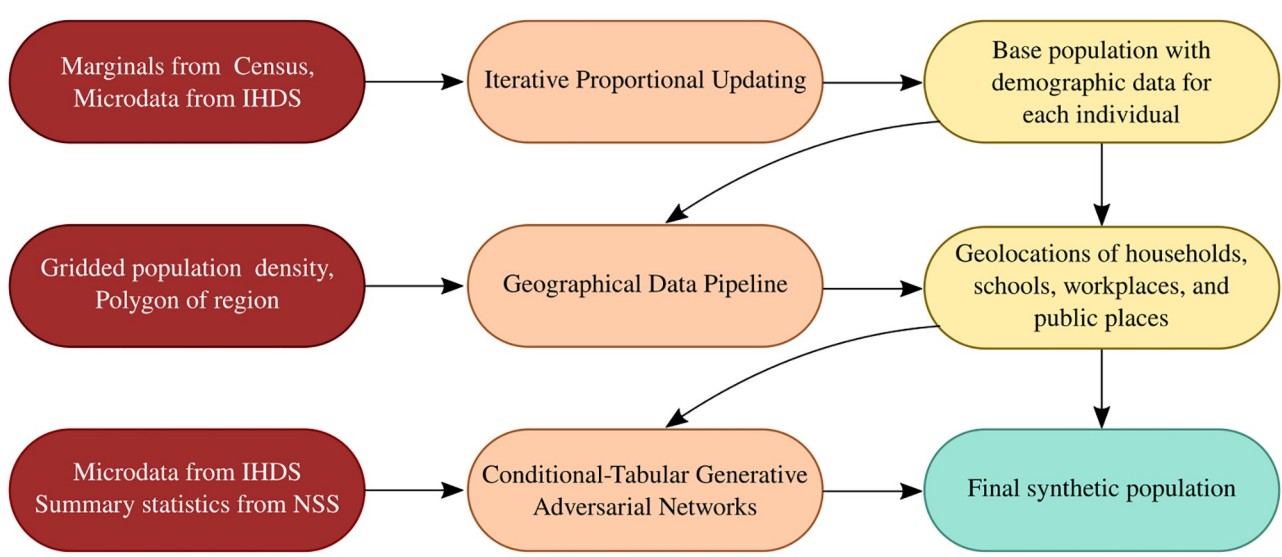

**Fig 1. Pipeline for synthetic population generation.** Census, gridded population, and survey data are first processed and subsequently used as the input to generate the final synthetic population.

and sex of individuals within a family, the geographical distribution of households, workplaces, and schools within administrative units (cities, districts, and states), and the number of people who are associated with a workplace or school. The population generation pipeline described below is illustrated in Fig 1.

Iterative Proportional Updating (IPU) [50] is a statistical sampling technique that accepts both information regarding marginals from the real population, and individual-level data from a sample of the real population. IPU is used to iteratively scale up the sample to generate a large-scale population, ensuring that the marginal distributions for both households and individuals in the synthetic population match with the known marginal distributions in the real population.

While IPU generates a large-scale population in which households have a realistic distribution of the number of family members and their age and sex, the joint distribution of various attributes in the synthetic population should also resemble those of the real population. To achieve this, we use a machine learning technique, Conditional Tabular Generative Adversarial Network (CTGAN) [51], to add individual-level attributes for which marginals are not available. CTGAN is trained using data from a sample survey and used to generate highly correlated individual attributes like heights, weights, and comorbities. This provides a reasonable joint distribution of various individual-level attributes. Additionally, different trained instances of our CTGAN model can also be used to combine different survey datasets with an overlapping set of attributes.

A realistic geographical distribution of the population must describe the spatial distribution of households, workplaces, schools, and public places, and the number of individuals that frequent them. We use grid population density data to assign individuals to specific locations. A row in the grid population density data consists of latitude ($X$), longitude ($Y$) and the number of individuals ($Z$) who live within a square of side 1 km, centred at ($X$, $Y$). We use rejection sampling to assign geo-location points within a geographical boundary, faithful to population density distributions [49]. An individual is then assigned locations such as workplaces, schools and public places, based on a probability inversely proportional to their distances from the individual's household [52].

The sampling method determines the geographical boundaries of the region for which data is being generated using a polygon provided by a geojson file. We filter the population density dataset by only retaining those rows for which the latitude $X$ and longitude $Y$ are within the boundary polygon of the region. We sample, with replacement, a large number of latitude-longitude pairs from this filtered subset, using the number of individuals ($Z$) as the weight for this sampling. For a given row in the sample, we then add uniformly distributed random noise to both the latitude $X$ and longitude $Y$. We reject those latitude-longitude pairs that are not within the polygon, until the required number of points is reached. Given that we do not have access to workplace distribution data, we make the natural assumption that it follows the population density distribution as well. Our technique can use actual workplace density data to obtain distributions more consistent with reality.

Individuals in our synthetic population are associated with external locations which include workplaces, schools, and public places. These locations define their contact networks. All individuals are assigned job descriptions. This job description is sampled with replacement from the list provided in the IHDS-II dataset. Individuals below the age of 3 are "Homebound". For individuals above the age of 3 and below the age of 18, "Student" is assigned as the job description. The number of workplaces, schools and public places varies for each district. Each workplace has three attributes: workplace type, latitude and longitude. Just as for the job description, the attribute "workplace type" is generated by sampling (with replacement) from the list of jobs in the IHDS-II subset. Specific workplaces are assigned as schools by looking at a subset of workplaces of individuals with job descriptions set to "Teacher". Students are assigned to schools and adults to workplaces, provided they are not homebound. We assume that agents are more likely to visit public places which are closer to their homes.

Given an individual with home latitude $X_h$ and longitude $Y_h$ and a list of $k$ possible external locations $\{E_k\}$ with latitude and longitude pairs $(X_{E_1}, Y_{E_1}), \ldots, (X_{E_k}, Y_{E_k})$, we calculate the Euclidean distance $r$ between the home of the individual and the external location. Among those who visit external locations, a fixed fraction is labelled as taking public transport. (Although we do not explore this in this paper, such a label would allow us to examine the possibility of increased transmission in such crowded locations. This has been examined in prior work [39]). We then choose a real-valued function $f(r)$ which is strictly decreasing and non-negative in the positive real domain, and weight the probability of an individual being assigned the external location by the value $f(r)$. For our analysis, we choose $f(r) \propto 1/r^2$ so that the probability of an external location being assigned is inversely proportional to the square of Euclidean distance between the individual's house and the external location. A geodesic distance metric can also be used, but the difference is insignificant at these scales.

As an example, in S2 Appendix we describe the creation of a synthetic population for Mumbai, along with a visualization of the geographic distribution of synthetic households, workplaces and schools in the combined districts of Mumbai and Mumbai Suburban. We also include a statistical analysis of the synthetic population vis-a-vis the survey population.

The code that generates the synthetic populations used in this paper available on the BharatSim GitHub page (www.github.com/bharatsim). In addition, the synthetic populations used in our simulations can be found on the BharatSim website (bharatsim.ashoka.edu.in).

One of the challenges in making a synthetic population is the need for up-to-date survey data. The most recent iteration of our main source of microdata, the IHDS-III survey, was expected to be released in 2023, but has still not yet made public. Additionally, the last India-wide census was conducted in 2011. The absence of up-to-date census data has made the task of extrapolating to the current date far more difficult. Following most, if not all such modelling efforts, we have used data from the 2011 census to generate the populations used in this paper.

However, we understand that some modellers may prefer to use a more up-to-date population. To facilitate this, we have used state-wide population projections from the Niti Ayog, India's apex planning body, extrapolating to 2021, and assuming that distributions remain similar in form. This projected population is also available from the BharatSim website.

**2.1.3 Population benchmarks.** We evaluate the synthetic population based on the following three criteria: (i) does the distribution of individual features in synthetic population match those in sample surveys? (ii) do the joint distributions of selected features in synthetic population match those in sample surveys? and, (iii) can our synthetic population replace survey data in a machine learning regression task?

S2 Appendix compares the marginal distributions of attributes of age, height, and weight across samples of the real and the synthetic population for the combined districts of Mumbai and Mumbai Suburban. We work with a randomly chosen subset comprising 10,000 individuals to compare with the survey data, although our synthetic population has 12 million individuals.

To benchmark the population, we compared the marginals of individual features (age, height, weight, sex) in the synthetic population with those in the survey data to quantify the similarities of their underlying statistical distributions. The results of our tests are shown in S2 Appendix.

To examine whether the synthetic data could reproduce correlations present in the survey data, we use a machine-learning regression task. We first attempt to predict the weight based on age, sex and height. Similarly, we attempt to predict the height based on age, sex, and weight. We use two machine-learning models: Linear Regression and Multi-layer Perceptron (MLP) Regression [53–55]. The results of these regression tasks are reported in Table 2. We begin by training each model on a subset of the survey data. We choose this subset to be 80% of the entire sample. We then report the accuracy of the model on the remaining 20% of the sample. Next, we instead train our models with the entire synthetic population, and again report the accuracy of the model on the same 20% of the survey data. The results of these comparisons are shown in Table 2.

## 2.2 The simulation engine

The simulation engine framework allows modellers to directly specify their models using a domain-specific, high-level language. This domain-specific language (DSL) is based on the *Scala* programming language [56], allowing modellers to extend their knowledge of *Scala* when creating their models. The DSL provides abstractions and constructs that make it easier for modellers to define general agent-based models of arbitrary complexity. It simplifies the creation of customized data structures, the execution of scenarios and experiments, as well as

**Table 2. Benchmarking synthetic population quality.** To assess the viability of using synthetic populations as substitutes for real populations, we analyse the predictive performance of various machine-learning models using both real and synthetic data. Each model is tasked with predicting weight (or height) based on age, sex, and height (or weight). Case 1: the model undergoes training and testing using survey data. Case 2: an identical model is trained on synthetic population data and tested on the survey data. This table lists the accuracy of each model in predicting outcomes for linear regression and MLP regression.

| Machine-learning model | Predicted attribute | Model accuracy | |
|---|---|---|---|
| | | Case 1 | Case 2 |
| Linear Regression | Height | 72% | 69% |
| | Weight | 78% | 76% |
| Multi-layer Perceptron Regression | Height | 69% | 65% |
| | Weight | 71% | 72% |

the integration with a synthetic population. Modellers can thus focus on the specific problem they wish to address even without being highly computationally skilled.

**Agents** The `Agent` class encapsulates an individual agent's attributes, behaviours, and relations with other agents or entities. Modellers can extend the framework's inbuilt `Agent` class as required, including behaviours specific to the context of the problem they are addressing. For example, the `StatefulAgent` described below is one such inbuilt extension of the `Agent` class.

**Behaviour** An agent's behaviour is defined as an action that is executed at every simulation timestep (or "tick"). For example, an agent might have some probability of getting themselves vaccinated at each time-step. This probability can depend on their attributes like their age, socio-economic status, and prior vaccination history.

**Stateful Agents** A `StatefulAgent` is a specialized extension of the `Agent` class that supports the framework's Finite State Machine (FSM). This is particularly useful in models where agents can exist at any given time in only one "state", but where transitions exist between these states. The `StatefulAgent` interface can define and control these transitions between states. These include actions that must be performed when agents enter or exit these states, as well as actions that are performed in every simulation tick.

**Networks** BharatSim is a network-based simulation model, where networks determine agent interactions. Every `Agent` or a `StatefulAgent` is a `Node` and a `Network` is composed of such `Nodes`, with `Relations` between them. The framework allows the modeller to define relations between agents that mimic real life network locations like houses, offices, restaurants, or other public places. These networks can, in principle, be dynamic.

**Relations** Different `Nodes` can be related to each other using framework-defined as well as extendable `Relations`. For example, an agent can be linked to his or her house, as well as his or her workplace. Two agents or workplaces can also be related to each other in this fashion.

**Schedules** Schedules define the `Network` locations of `Agents` at a given instant in the simulation. For example, the `Schedule` can specify that a specific `Agent` is part of her "House" network for 0 to 8 simulation ticks and "Office" network for 9 to 18 ticks. The BharatSim DSL allows for the definition of a custom `Schedule` using predefined units (`Day`, `Week`, `Month`), but also allows modellers to define custom units of time, as per their requirement.

**Interventions** Interventions are a set of externally defined rules imposed on the agents, such as lockdowns or vaccination drives. BharatSim allows for the incorporation of such interventions in order to study a range of counterfactual scenarios. Modellers can define multiple types of interventions, with different activation, deactivation, and reactivation conditions over the course of the simulation.

**Simulation** In BharatSim, model developers have to define a `Simulation` in order to be able to specify input data, register agents, define schedules, and generate output. Along with monitoring clock ticks, the `Simulation` stores all common data in a `Context`, which is shared globally and from which other objects like agents can query information.

### 2.2.1 Creating `Agents` and `Networks`.

The simulation engine accepts the synthetic population as a CSV file. This data is stored in a *graph* data structure, a network of nodes

related to each other through user-defined relations. These nodes represent both individual agents as well as network locations such as households or offices. The framework defines a `Node` class which allows for relations to be established between other nodes. Using such a data structure makes the framework flexible and domain-independent. This can be implemented in one of two ways, either by using Neo4j, a graph database, or using the *Scala* programming language's scalable map implementation, `TrieMap`. Both these structures optimize data operations, allowing the simulation to scale efficiently to larger populations.

*Scala* allows programmers to define their own datatypes, justifying its choice for the BharatSim simulation framework. In particular, a user can define a special "class", a user-defined blueprint with a set of properties or functions that are common to all objects of this datatype. The user could then create different objects as "instances" of this datatype, just as a user might create different variables to be instances of the `Integer` datatype. Additionally, these classes can then be "extended" to define new datatypes. These new datatypes "inherit" the attributes and methods of their parent class, but may also possess new attributes and methods of their own.

The framework uses the `Node` class to define the `Agent` and `Network` classes. The `Agent` and `Network` classes can be further extended by the modeller to define the individual agents and locations of their own model. The `Agent` class possesses a `addBehaviour` function which can be used to define agent behaviours.

In Scala, `case` classes are particularly useful for modelling immutable data. In the models specified below, the agents and network locations are thus defined as `case` classes that extend these inbuilt framework-defined classes. Additionally, BharatSim implements a Finite State Machine (FSM); with this implementation, a modeller may define classes that extend the inbuilt `State` trait to distinguish agents that exist in different disease states. In order to use this FSM, the framework extends the `Agent` class to define `StatefulAgent`, an agent that can be in only one `State` at any given instant of time. The `State` class possesses two inbuilt functions:

a. An `enterAction`: a function that defines an action that is to be performed only whenever an agent enters this particular state, and

b. A `perTickAction`: a function that defines an action that is repeated by an agent at every time-step (or "tick") for as long as they are within this state.

Additionally, every extension of the `State` class also possesses an `addTransition` function which allows the modeller to define (i) which `State` a `StatefulAgent` can transition to from the current `State`, and (ii) under what conditions such a transition can occur. Very much like the `addBehaviour` function, the `addTransition` function is run for every agent at every time-step, to check whether a transition between states should occur.

Agents are assumed to move between different network locations to interact with each other. These locations can be modelled by extending the inbuilt `Network` class. Just as with the `Agent` class, these classes too may possess an arbitrary number of attributes. Additionally, the `Network` class has an inbuilt `getContactProbability` function which can be used to define, for example, a location dependent relative risk of infection.

Every individual agent follows a schedule that is defined by the modeller. Such schedules specify agent locations across time. These schedules can be dynamic, can depend on the current state of the agent, and can be affected by interventions that are imposed. In addition to the predefined units of time (`Hour`, `Day`, `Week`, etc). schedules may also be linked to user-defined units of time (instances of the `ScheduleUnit` class). In BharatSim, these schedules can be registered using the `registerSchedules` function. When schedules are registered,

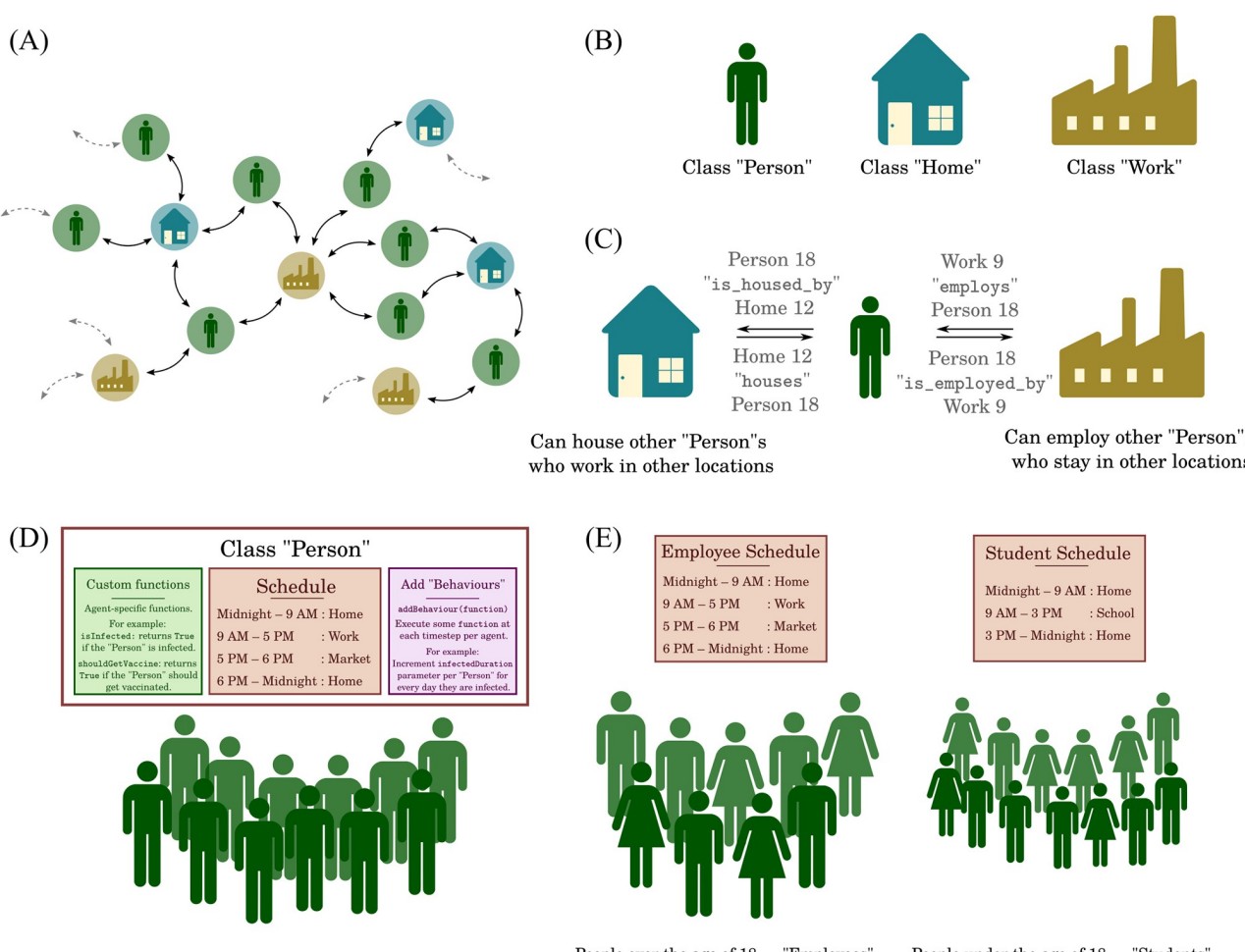

**Fig 2. Components of the simulation engine.** (A) A schematic of a graph between nodes representing agents, homes, and work locations. (B) These nodes are user-defined extensions of the in-built `Node` class. (C) The graph is created by creating bi-directional relationships between these user-defined classes. (D) Agents can have user-defined schedules using user-defined units of time. Additionally, the agent class may posses custom user-defined functions. Some of these functions, called "behaviours", can be set so that they are executed by every agent at every time-step. (E) Agents' schedules can be customized based on attributes from the synthetic population like their age, job label, and socioeconomic status.

a priority can also be assigned to them. This may be used to give a lockdown-schedule a higher priority than the routine daily schedule of any agent. In Fig 2 we show a schematic of some of the different components of the simulation engine described above.

**2.2.2 Defining disease dynamics across `States`.**   The movement of individual agents between different states can be modelled using either transitions or behaviours, depending on whether we use the inbuilt finite state machine or not. If an individual agent is susceptible, then we decide whether or not they may become infected using a probability that depends on the number of infected individuals they are in contact with in a given interval of time.

When an agent transitions into any other disease state, an "exit-time" is chosen, based on the current time and the probability distribution associated with that disease state. The modeller can use the inbuilt random number generators in *Scala* to define and use these distributions. If an exiting agent has multiple possible states to transition into, one of these states is chosen based on the disease parameters. The individual then stays in this disease state until the current time exceeds their exit-time, at which point a further transition occurs.

**2.2.3 Running a simulation.**   For a typical simulation we define a `Person` class as an extension of a `StatefulAgent`, in addition to two extensions of the `Network` class: `Home` which represents the agent's household, and `Office`, which represents their workplace. The disease states of our model are defined as extensions of the `State` trait.

Once these basic components are defined, we create the `Main` class of our simulation. Within this, we create a new instance of the framework's `Simulation` class. The synthetic population is then imported using the `ingestData` function of the `Simulation` class. This is done using the framework-defined `ingestCSVData` function. This function accepts a CSV file as well as a user-defined "mapping" function which creates a graph. At the end of the population import, a graph is created, containing all the individual agents and their attributes, as well as all the home and work locations in our simulation.

During the population import process, relations are established between the different nodes. Relations are single-directional links between two nodes, distinguished by a string label. For example, a `Person` and `Home` are linked by the "`STAYS_AT`" relation. Similarly, a `Home` "`HOUSES`" a `Person`. Relations can be established between any two nodes, for example two `Home` nodes can be related using a "`NEIGHBOUR`" relation, and so on.

Once the synthetic population has been imported and the graph created, we call the `defineSimulation` function of the `Simulation` class. Within this, we (i) register the agents in our simulation using `registerAgent` function, and (ii) register the different disease states in our simulation using the `registerState` function (if we are using the inbuilt FSM).

We can then define schedules that the individual agents are required to follow, based on their attributes (age, employment status, and so on). Once the schedules have been defined, we can also register an action to be performed based on user-defined conditions given to the `registerAction` function. For example, one could choose to call the predefined `StopSimulation` action (which ends the simulation) when the number of infected agents in the population is zero.

The modeller can also create a new instance of the `CsvOutputGenerator` class and send it to the `register` function of the `SimulationListenerRegistry`. This function is called at every time step and uses the `CsvOutputGenerator` to write a single file to an output file. This can be used to store, for example, the aggregate numbers of susceptible, infected, and recovered individuals in our population. The modeller can use the `onCompleteSimulation` function of the `Simulation` to run specific commands when the entire simulation has finished. This function should also end with the `teardown` function which clears the data and closes the connection with the database.

Further details, including a step-by-step guide of how to create a simple agent-based SIR model using BharatSim, can be found in the "Writing your first program" section of the BharatSim documentation.

## 2.3 The visualization engine

Using the simulation engine, the modeller can write simulation data to an output file. Such a file could contain, for example, the number of individuals in different disease states for every time step, or more specific geo-spatial information. The visualization engine can then read this output and create multiple dashboards with different types of graphs that can help to analyse the results and visualize them.

The visualisation engine is self-contained, and can accept and visualize any CSV data file. Each dashboard provides the user with a combination of different graphs and charts, including line-graphs, histograms, and pie-charts. Additionally, the visualization engine can also

represent GIS data in the geoJSON format to plot heatmaps and choropleths. These can either be static, or can change with time based on the data provided in the input CSV file.

The visualization engine provides the following features:

**Data import** The visualization engine can import data in ZIP, CSV, and GeoJSON formats. The user can add, delete, and link files from different dashboards and can also add columns with custom formulae.

**Charts** The user can represent time-series data as a line-graph. Additionally, bar charts, histograms, and pie charts can be used to study aggregated data. Geographical information and data can also be visualized. Spatial variations in data can be visualized using either heatmaps or choropleths. Heatmaps visualize data in the form of "hot" or "cold" spots, with a warmto-cool colour scheme. Choropleth maps can also be used to visualize how a quantities vary across a fixed geographic area like a district or state, while simultaneously showing the extent of variation within a region.

**Project & dashboard management** Users can create a Project which can have multiple individual dashboards. Users can add, edit, and delete projects and dashboards for better management. Dashboards are auto-saved to avoid data loss. Users can easily add, edit, duplicate, and delete widgets and charts in each dashboards. Dashboards can be duplicated to allow users to duplicate certain design choices and widget configurations. Each widget also allows the users to export the output either as raster or vector data using the PNG and SVG file formats respectively.

## 2.4 Implementing a disease progression

The BharatSim framework allows a modeller to implement a variety of agent-based models. The disease states in these models can be labelled according to the compartments in associated compartmental models. In this paper, we restrict ourselves to the compartments described in the INDSCI-SIM model of Ref [42]. This model uses the disease progression shown in Fig 3. It has been used to describe the disease trajectory across the first wave of COVID-19 in India. However, BharatSim allows users to easily use different disease progressions. Indeed, in our study of mpox, described in detail in S9 Appendix, we have modelled a different set of disease compartments.

The INDSCI-SIM model assumes 9 compartments: Susceptible ($S$), Exposed ($E$), Asymptomatic ($A$), Presymptomatic ($P$), Mildly Infected ($MI$), Severely Infected ($SI$), Hospitalised ($H$), Recovered ($R$), and Dead ($D$). Those in the Susceptible compartment can be infected by all individuals in any infectious compartment, whereupon they transition to the exposed compartment. Exposed individuals can either become asymptomatic, exhibiting no symptoms and eventually recovering, or presymptomatic, wherein they transition into either the mildly or severely infected compartments. Mildly infected individuals recover, while severely infected ones are eventually hospitalised. They go on to either recover or die. Extensions of the INDSCI-SIM model that incorporate vaccinations have been used to describe subsequent waves in the country [57].

The synthetic population described in Section 2.1 provides us with individual-level information about the age distribution, household size distribution and composition, and comorbidities. This information provides further granularity to the description of disease states and the transitions between them. In the model discussed here, for example, the individual agent's probability of branching between the asymptomatic and presymptomatic states is age-

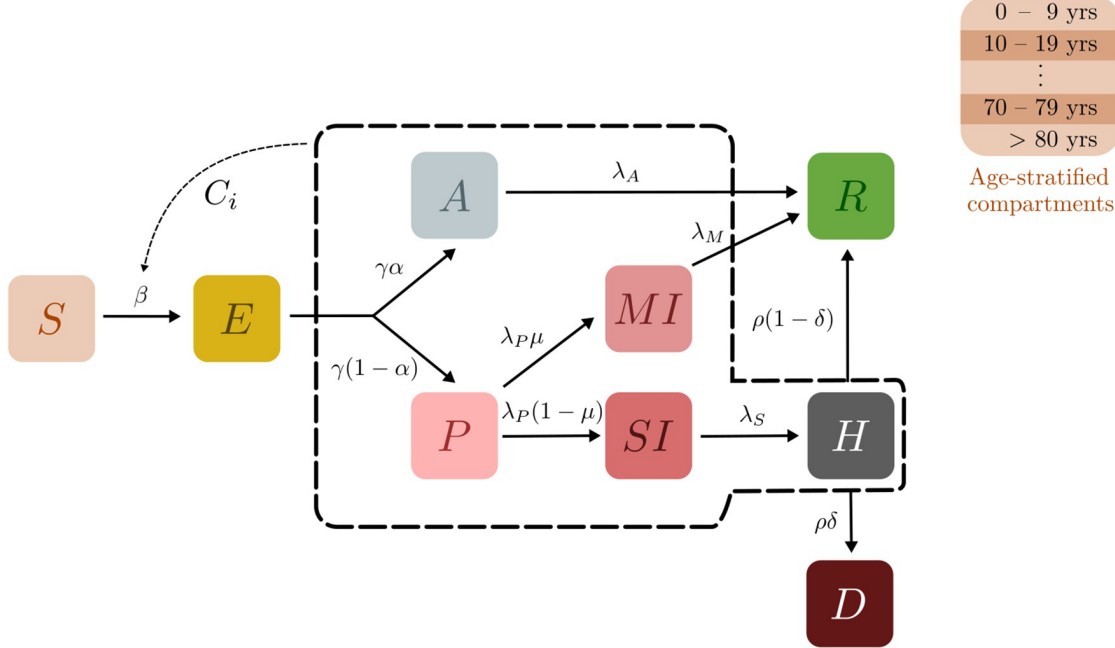

**Fig 3. Schematic of the epidemiological model.** Boxes show the 9 compartments of the model, i.e., the set of states that characterise each individual in the population. The compartments are (S)usceptible, (E)xposed, (A)symptomatic, (P)resymptomatic, Mildly Infected (MI), Severely Infected (SI), (H)ospitalised, (R)ecovered, and (D)ead. Black arrows indicate possible transitions between states. The dotted lines indicate that the rate of infection of Susceptible individuals is an increasing function of the number of infected individuals they are in contact with. Recovered individuals are assumed to be immune to further infection in this model, however the possibility of including a non-zero recovery rate has been implemented.

dependent, with older individuals more likely to develop an acute case of the illness. We do the same for the other branching probabilities, as shown in Table 3. In addition to the agent's age, the synthetic population also provides us with the coordinates for agents' homes and work-places. No other columns from the population are currently used, although it is straightforward to extend our simulation to incorporate, for example, the effect of comorbidities on disease spread.

The residence times that different individuals spend in different compartments can, in principle, also be age-stratified. For simplicity here, we assume these parameters to be the same across all age-groups. BharatSim allows the modeller the possibility of choosing these residence times from different probability distributions including the exponential distribution, as simple compartmental models assume, or the lognormal distribution which we use in this paper on grounds of its wider applicability. A set of convenient parameters for these different

**Table 3. Branching ratios for different compartments.** In our model, we assume that an individual agent's probability of branching between different compartments is age-stratified. Individuals are divided into age-bands of one decade each. When an individual agent exits a compartment that branches into multiple possible compartments, they are assigned a compartment based on this age-stratified probability. These rates have been taken from Ref [59].

| Parameter | 0–9 | 10–19 | 20–29 | 30–39 | 40–49 | 50–59 | 60–69 | 70–79 | 80–89 | 90+ |
|---|---|---|---|---|---|---|---|---|---|---|
| Relative Susceptibility | 1.00 | 1.00 | 1.00 | 1.00 | 1.00 | 1.00 | 1.00 | 1.00 | 1.00 | 1.00 |
| Asymptomatic Probability | 0.50 | 0.45 | 0.40 | 0.35 | 0.30 | 0.25 | 0.20 | 0.15 | 0.10 | 0.10 |
| Mild Infection Probability | 0.999 | 0.997 | 0.988 | 0.968 | 0.951 | 0.898 | 0.834 | 0.757 | 0.727 | 0.727 |
| Probability of Death | 0.0185 | 0.0187 | 0.0143 | 0.0166 | 0.0340 | 0.0500 | 0.0970 | 0.2100 | 0.2200 | 0.2200 |

**Table 4. Residence times in different compartments.** When an individual agent enters a disease state, they remain in this compartment for a period of time drawn from the probability distributions given above [58].

| Parameter | Description | Distribution |
|---|---|---|
| $\tau_E$ | Exposed-to-infectious duration | Lognormal(4.5,1.5) |
| $\tau_{\text{sym}}$ | Time to onset of symptoms | Lognormal(1.1,0.9) |
| $\tau_A$ | Recovery time for Asymptomatics | Lognormal(8,2) |
| $\tau_{MI}$ | Recovery time for Mild cases | Lognormal(8,2) |
| $\tau_{SI}$ | Time before hospitalization | Lognormal(1.5,2) |
| $\tau_H$ | Time spent hospitalised | Lognormal(18.1,6.3) |

distributions can be obtained from Ref [58], and can be found in Table 4. These can be updated as more information becomes available.

We also incorporate the possibility of vaccination drives in our model. Vaccines are assumed to reduce both the relative risk of infection as well as the probability of symptomatic disease. We also assume that vaccinated individuals are less likely to transmit the disease, as shown in Ref [60]. The vaccine's efficacy is assumed to ramp-up to its maximum value over a period of 14 days. Additionally, the dose-interval is currently set to 90 days between doses; see S5 Appendix for more information.

All individuals above the age of 20 are assumed to be eligible to receive vaccines, provided they are not exhibiting symptoms. We can begin our simulation with either a vaccine-naive population, or a population where a certain fraction has already received partial protection through one or two shots of the vaccine. For the simulations discussed here, these numbers are chosen to be 20% and 10% respectively, to match the situation in India as of August 1, 2021. These initial vaccines are distributed as follows: first, 80% of the individuals above the age of 60 are assumed to be vaccinated by the start of the simulation. Next, any remaining vaccines are distributed randomly among the full (age-stratified) eligible population.

Additionally, vaccines can be rolled out in different phases, targeting eligible individuals of different age-groups. In the simulations described below, we include two phases. In the first phase, individuals above the age of 40 are given first preference to receive vaccines. This phase spans one month. In case there are no eligible agents above the age of 40, the vaccines are randomly distributed among the entire population to avoid allotment-related wastage. In the second phase, all agents above the age of 20 (not currently exhibiting symptoms) are targeted for vaccination.

## 2.5 Network dynamics

An individual's movements in a day are defined through a "schedule" (see Section 2.2), specific to each individual. This schedule dictates which network location (home or workplace) the individual participates in at any given time. The infection spreads stochastically between individuals who share a location at any given time, at rates determined by the infection parameters as well as the vaccination status of the different individuals present in that location. In our simulations, we take a single time step to be 12 hours, defining the probabilities accordingly. Individuals move back and forth between their homes and workplaces or schools twice a day.

Schools are a specific subset of workplaces to which all school-going individuals are assigned. Schools consist of classrooms, and students are assigned specific classrooms and do not mix with students in other classrooms. Multiple network and school structures can be examined. For example, one may allow for students and/or teachers to move between the classrooms of the school. Classrooms may also be defined according to age brackets.

The force of infection $\mathcal{F}$ at any location $l$, the rate at which a single susceptible person becomes infected, is proportional to the product of the number of infectious people at that location and the infectivity parameter $\beta$. This is divided by a normalization factor $\mathcal{V}_l$:

$$\mathcal{F}(l) = \frac{\beta}{\mathcal{V}_l} \sum_{\substack{\text{agent } i \\ \text{in } l}} V_{\text{tr}}(i)\, C_q(i), \tag{1}$$

where $V_{\text{tr}}(i)$ is the factor by which the agent $i$'s probability of transmission is changed due to vaccination, $C_q(i)$ is an "isolation factor" (chosen in the range $0^+$ to 1). In our simulations below, this factor is 0.1 if the agent is hospitalised or quarantined and 1 if not. For density-dependent transmission, as in this paper, we can chose $\mathcal{V}_l$ to be the effective number of individuals in location $l$, not counting dead individuals and only a fraction of hospitalised individuals, as in [61]. However, this can easily be generalised to other types of transmission, including frequency-dependent transmission.

The force of infection at each location controls the probability $p_j$ that a susceptible individual $j$ in that location contracts the disease and therefore transitions to the exposed state in some time-step $\Delta t$:

$$p_j = V_{\text{inf}}(j) \times \mathcal{F}(l) \times \Delta t, \tag{2}$$

where $V_{\text{inf}}(j)$ is the relative risk of infection, given by the vaccination status of the individual $j$.

When an individual enters the exposed compartment or any of the subsequent compartments, a "sojourn-time" is drawn and stored from the distributions shown in Table 4. The individual remains in this compartment for this period. Once this time has elapsed, the individual transitions to the next state in the disease progression. If two possible outcomes exist—for example, suppose they can transition to both the asymptomatic or presymptomatic states—then one of the outcomes is chosen randomly, weighted by the appropriate (age-stratified) branching probability for that transition. For the new state, a new "sojourn-time" is drawn and stored.

Following an update of all agents, the time is incremented by $\Delta t$, all individuals are moved along the network according to their specified schedules, and the process repeated for the next time-step.

## 2.6 The models and underlying paradigms

There are two ways to model the disease progression in BharatSim and the modeller may choose between then. Both work with the same disease progression and synthetic population and implement the same interventions. In the first, the framework's inbuilt Finite-State Machine (FSM) controls the movement of agents between the different disease states. The framework ensures that an individual `StatefulAgent` can only possess one `activeState` at any given instant of time. Such an implementation is computationally efficient and simple to debug.

When we wish to model situations where multiple strains of a disease could coexist, such an FSM cannot be implemented without a proliferation of states. Indeed, an individual could have recovered from one strain, only to be infected with another, or might in principle be simultaneously infected with both strains. In the second implementation, we do not use the `StatefulAgent` class, but instead work with the `Agent` class. In this case, the individual's state is defined by a set of two attributes: `s1_disease_state` and `s2_disease_state`, which contain information of which states they are currently in with respect to the disease dynamics of each individual strain. In the model described here, being infected by one strain

reduces an individual's relative risk of infection to the second strain. We vary this relative protection from the second strain to see how varying levels of immunity provided by the first strain change the disease trajectory. (Reinfections for each individual strain are currently not implemented in the examples discussed below, although it is easy to include them.)

## 2.7 Interventions and complex dynamics

**2.7.1 Lockdowns and school-closures.** Non-pharmaceutical interventions like lockdowns or school-closures can also be implemented in BharatSim. For a simulated lockdown, individual agents' schedules are modified so that they remain at home. All agents are assumed to follow this rule, except those that are designated as essential-workers. A tunable fraction may additionally have a low adherence to policy-level interventions. School-closures work in a similar fashion: all students and teachers assigned to a school stay at home, depending on their level of adherence.

**2.7.2 Vaccination strategies.** A vaccination drive can be modelled as one of many types of `Interventions` in the BharatSim framework. Here, we extend the INDSCI-SIM model to incorporate a two-shot vaccination drive [57, 62]. We allow for a fixed daily vaccination rate as some fraction of the total population (note that this is not a fraction of the eligible population). Thus, with a total population of 10,000, a daily vaccination rate of 0.2% would mean that there are 20 vaccines that can be distributed among the eligible population every day. This number can be varied, so as to compare different allocations of vaccine doses corresponding to 0.0, 0.2, 0.4, and 0.8% of the population. Peak vaccination levels in India corresponded to roughly 0.4%. Because our model is age-structured, we can prioritize different age groups [63], and also allow for a fraction of the population to be vaccinated at the start of the simulation.

Vaccines are modelled through adjusting individuals agents' susceptibility to infection ($V_{\mathrm{inf}}$) and probability of developing symptoms after being infected (which is controlled by the age-stratified branching parameter $\alpha$, the probability of an individual being asymptomatic) [64]. Both of these changes cause a reduction in the overall probability of severe disease, hospitalization, and death. Vaccines are assumed to also reduce transmissibility by a factor $V_{\mathrm{tr}}$. In principle, disease severity could be a function of other features captured in the synthetic population, such as individual co-morbidities. For simplicity, we ignore this possibility here.

Our models further allow us to implement a two-dose vaccination strategy [65], with dose prioritization: the maximum protection that the first-dose provides is attained through a linear ramp-up over some period $T_{\mathrm{ramp}}$. Once this value is attained, the parameters that vaccination affects remain unchanged for some "inter-dose" interval $T_{\mathrm{id}}$ until the vaccinated individual becomes eligible for (and receives) a second dose, following which their protection is increased through another linear ramp-up.

Over the course of the simulation, vaccinations are administered at the end of every day. Dose prioritization is done by first allocating a certain fraction of the vaccines as "second-dose" vaccines. These will be used to vaccinate any agents eligible for a second dose on that particular day. Any remaining doses are distributed as first doses. To make this concrete consider a vaccine drive that begins on day 0, with daily vaccination rate of 1%, and an inter-dose interval of 90 days. We further assume a dose-prioritization of 80:20, that is, 20% of the vaccines are set assigned for second-doses while the remainder are used for first-doses. For the first 90 days, no individuals are eligible for second-doses. As a result, all the vaccines are administered to individuals requiring first-doses. On day 90, the first 10 individuals who received their first dose on day 0 are now eligible for second-dose vaccination. Thus, of the 20 vaccines set aside, 10 are used to get them fully vaccinated. The remaining 10 (as well as the assigned 80) are used as first-dose vaccines. In a similar vein, if any vaccines that are set aside

for first-doses and unused are used as second-doses where possible. This procedure ensures that no vaccines are wasted while eligible people remain in the population. See S6 Appendix for more information.

In this paper, for illustrative purposes, we have chosen a vaccination drive that is rolled-out in two phases. For the first 30 days, individuals older than 40 years of age are prioritised for vaccines. After this period, vaccines are provided to all individuals over the age of 20. Any excess vaccines are then distributed randomly among the remaining adult population, when possible, thus ensuring no wastage. We also ensure that 80% of the initial vaccinated population consists of individuals above the age of 60 years. We set $V_{tr}$ (the reduction in transmissibility) to be 40%. The linear ramp-up period over which the maximum vaccine efficacy is attained and the minimum duration between doses are set to $T_{ramp}$ = 14 days and $T_{id}$ = 90 days, respectively. The doses are prioritized in a ratio of 80:20, with 20% of the doses being set aside as second-doses, whenever eligible agents are available.

**2.7.3 The impact of school-reopenings.** We consider circumstances under which governments may consider allowing schools to be reopened in the background of a pandemic. To do this, we construct communities that simulate the catchment area for a given school. This community is chosen to be a section of the synthetic population consisting of 20,316 individuals and with 6500 homes, 120 workplaces, and 1 school. Of the individual-level attributes included in the synthetic population, we restrict ourselves to the age, and the home, work, and school locations of the individuals.

In the school, individuals are assigned to a specific classroom, and no movement is allowed between classrooms. Thus, in our simulations the relevant parameter is the number (and consequently the density) in the classrooms. We assume 100 classrooms per school. S7 Appendix shows the distribution of home sizes, workplace sizes, and classroom sizes in our model community, along with the age distribution. A certain fraction of the individuals in the population (30% in our case) is homebound.

**2.7.4 Introducing multiple strains.** Infections with different variants follow the same disease progression indicated in Fig 3, with different transition parameters. Here we assume that the two strains differ only in their value of $\beta$. The first strain is considered to be less transmissible, i.e. with a lower value of $\beta$, than the second one. All other parameters are identical, including the residence times for compartments and the branching ratios.

In addition, we consider the possibility of the first less-transmissible strain providing an agent with some immunity to infection from the second strain [66]. It is thus more likely that the second strain will infect someone who has escaped infection from the first strain. We vary this likelihood, as well as levels of vaccination [67], to explore counterfactual scenarios. Additionally, we also assume that an agent who has contracted the first strain is more likely to be an asymptomatic carrier of second strain than a symptomatic one, and vice versa. The second (more transmissible) strain is seeded with 10 individuals after the first strain has nearly ended. However, our simulation allows for both strains to be seeded simultaneously and to coexist as well [68–70].

## 3 Results

### 3.1 Simulating disease spread in a model Indian city

We use our model to simulate the city of Pune, with a population of 3.13 million agents, exploring the different disease trajectories induced by pharmaceutical and non-pharmaceutical interventions.

We consider first the effect of a lockdown on the disease trajectory. Fig 4A shows the fraction of recovered over all age-groups with time in different scenarios. In these scenarios, the daily vaccination rate is varied, and 15-day lockdowns are introduced when the total number

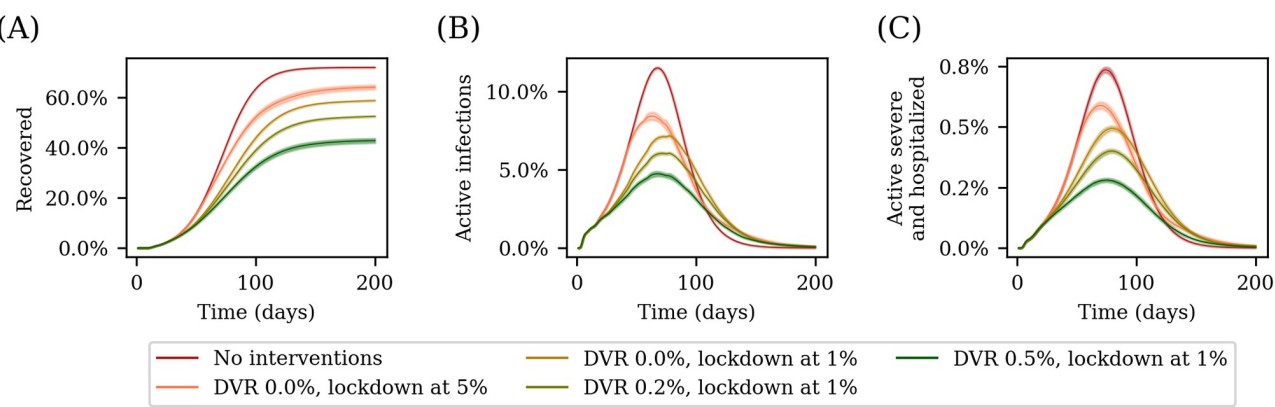

**Fig 4. Simulating disease trajectories with different interventions.** The plots show the disease trajectory for the city of Pune in five different scenarios: without any interventions, with 15-day lockdowns beginning when the number of active infections is 1% and 5% of the population respectively, with a lockdown starting with 1% active infections and concurrent vaccination drive at two different vaccination rates of 0.2% and 0.5%. (A) shows the number of recovered as a function of time, (B) the number of active cases as a function of time, and (C) the number of active severe cases and hospitalizations. We see that imposing a lockdown reduces the total number of people who contract the disease, an effect that is enhanced by starting the lockdown earlier. A concurrent vaccination drive with a high vaccination rate can further augment this. In all scenarios with a vaccination drive, the drive begins on day 0. The curves are averages over 25 runs, with the error bars corresponding to 1.96σ.

of infections crosses a threshold of either 1% of 5% of the total population. If left unchecked the pandemic would have led to 70% of the population having contracted the disease during the epidemic. With the introduction of a 15-day lockdown that begins when the number of active cases is 5% of the total population, this number reduces to close to 60%. The number further reduces if the lockdown began even earlier (when the number of active cases is 1% of the population).

We next consider the role of vaccinations. We begin a lockdown when the number of active infections is 1% of the population, and vary the daily vaccination rate of the vaccination drive. By comparing these scenarios with the curve representing the case with the lockdown but no vaccination drive in Fig 4A and 4B, we see that an increased daily vaccination rate reduces peak of the infection. The number of severe and hospitalised, shown in Fig 4C, follows the active-infections curve, although lagging slightly behind it in all scenarios.

Additionally, we also study what effect the initial spatial configuration of the disease has on its trajectory. In Fig 5, we compare two such scenarios: in the first, a total of 3000 individuals (roughly 0.1% of the total population of Pune) are seeded in a single ward, and the disease is allowed to spread. In the second scenario, the same number of individuals are chosen from a much smaller geographical region (a collection of 750 homes), and the disease is again allowed to spread. The figure compares the spread of the disease due to these two initial conditions.

Finally, we also compare the effects of different lockdown strategies on disease dynamics. In particular, we study the effect that locking down the most populous wards of the city has on the spread of the disease. In order to do this, the following procedure is used: the entire city is divided into a grid whose unit cell is 200 m × 200 m. Every five days, we count the number of infected individuals in each of these grid boxes. An individual is assumed to count towards the number of infected in a grid box if their household falls within its geographical boundaries. We place the most populous central wards under a lockdown for (i) 15 days and (ii) 30 days when the number of cases is 1% of the total population, prohibiting all entry and exit from these wards, and observe the effect this has on the disease dynamics. In Fig 6 we show the results for one such run in which we see that locking down the most populous wards can slow down the spread of the disease in the city, thereby flattening the curve of infections.

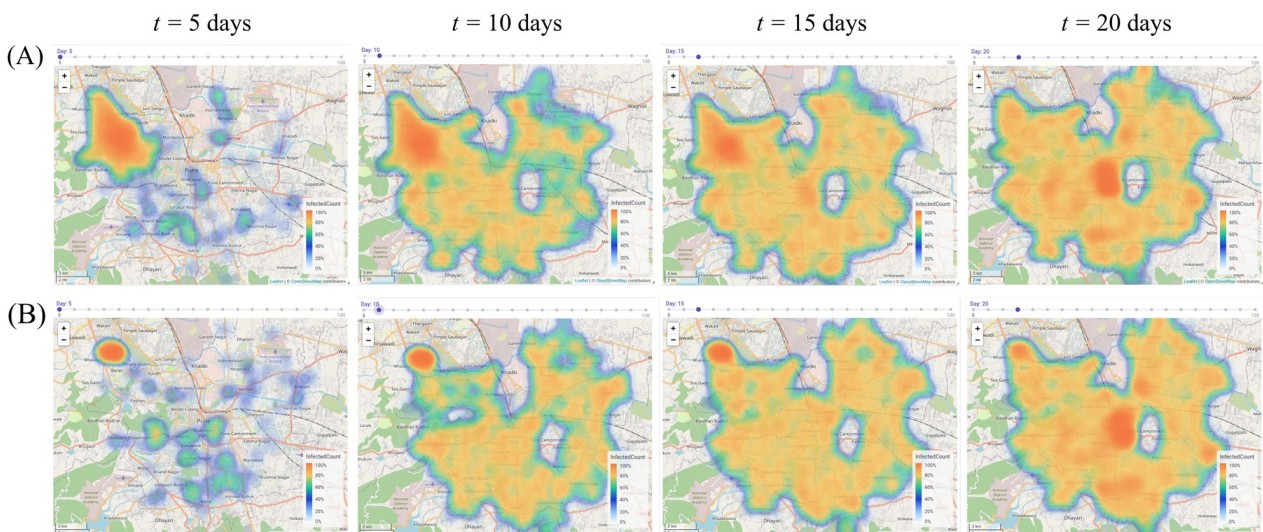

**Fig 5. Effect of initial infection seed on disease spread.** The heatmaps represent the density of the infected at different instants of time for two scenarios. In each case, a single simulation with an initial infection seed of 0.1% of the total population was seeded in (A) uniformly across a single ward, and (B) a geographical cluster of 750 families in a single ward. This illustrates how BharatSim can be used to model the spatial hetereogeneity in disease spread. The central ellipse that remains unchanged across time represents the Army Cantonment region for which no publicly-available data exists. The map image is produced by the BharatSim visualization engine with base maps obtained from OpenStreetMap (https://www.openstreetmap.org/).

In reality, the south-west of Pune saw a larger number of cases than other regions. This has been attributed to the relative importance of public transportation, variations in adherence to non-pharmaceutical interventions and inhomogeneities in case-records, among other explanations.

We recognise the utility of at least an approximate comparison to real data. We have chosen to use our model to describe the onset of the COVID-19 pandemic in the city of Pune, comparing model results to real data. We choose a time period between the 1st of March and the 15th of July, 2020 for this comparison for the following reasons: (i) the simulations are initiated when the lockdown started so spread would primarily have been driven by in-house transmission, (ii) the virus dynamics was largely dominated by a single variant, and (iii) we can study the impact of the relaxation of the nation-wide lockdown on cases.

This direct comparison is shown in S8 Appendix. Briefly, our results are consistent with (i) a relaxation of the first lockdown leading to an increase in cases, (ii) a large background of undetected infection, arising mainly from asymptomatic cases, (iii) an indication that the epidemic arising from the initial strain had run its course by the end of the second lockdown and that (iv) that a complex mix of more transmissible variants would have determined the later-time course of the disease. These agree with a retrospective analysis [59, 72].

Extending our study beyond this window of study would have involved accounting for the emergence and dominance of multiple such variants, as well as an ever-changing suite of non-pharmaceutical interventions. We will leave these to later work. Our model here is not intended to precisely reproduce these complexities, but to describe a framework within which such questions can even be addressed.

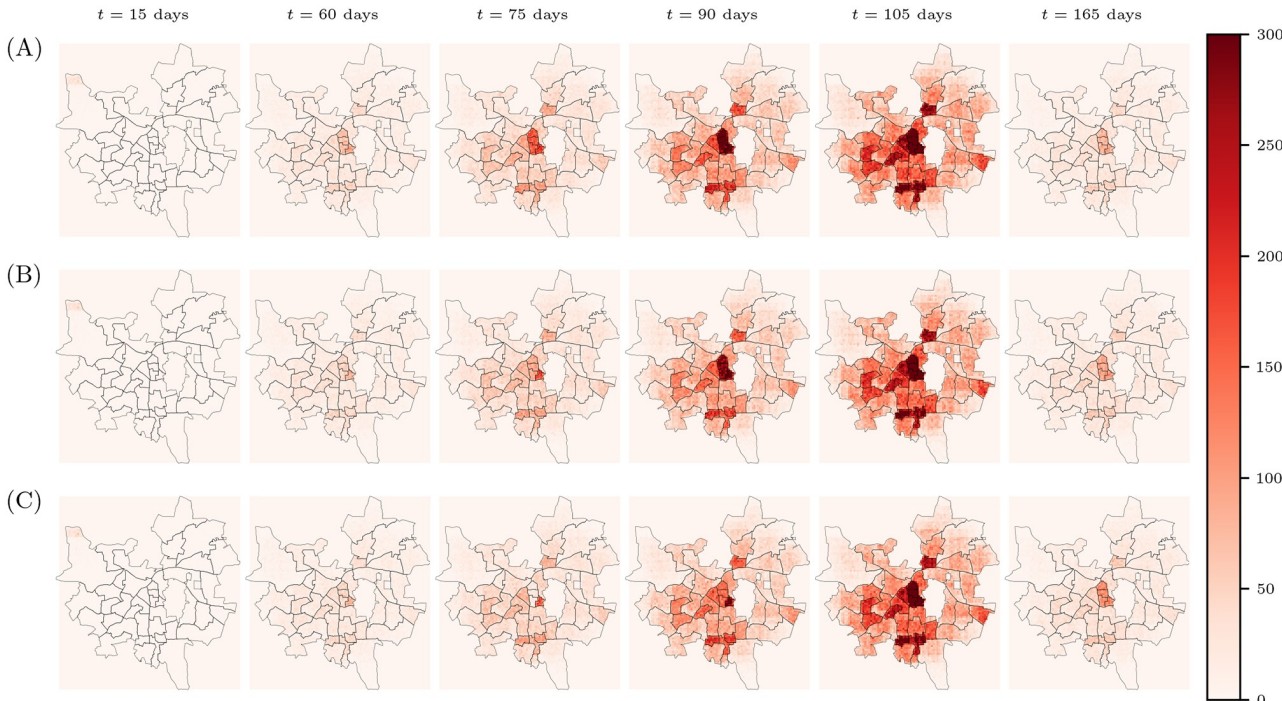

**Fig 6. Effects of different lockdown strategies.** The city of Pune is divided into a grid of 200 m × 200 m, and the number of infected individuals with households in each grid box is plotted as a heatmap. In scenario (A) no interventions are applied and the disease is allowed to progress freely through the population. In order to test the effect that locking down the most populous wards has, these wards are locked down for a period of (B) 15 days (C) and 30 days when the number of active infections is 1% of the total population, which corresponds to day 50. From these graphs, we conclude that locking down the most populous wards can slow down the spread of the disease in the city, thereby flattening the curve of infections. The underlying map of Pune is provided by the Spatial Data of Municipalities (Maps) Project by Data{Meet} [71], made available under the Creative Commons Attribution 4.0 International license. As before, the central region that remains unchanged represents the Army Cantonment area for which we have no data.

## 3.2 Effects of school reopening on the disease trajectory

To study the impact of school reopening on the disease trajectory, we simulate a number of different counterfactual scenarios. In these simulations, we assume schools are initially closed (i.e., children stay at home) as was the case in India for a close-to two-year period since March 2020. We then examine the effect of opening schools at various times during the epidemic. We assume 20% of our population is initially vaccinated, with 10% having received the second-dose. We further assume that vaccinations proceed at a constant daily vaccination rate with an appropriately chosen prioritization of first and second doses.

We consider two different initial conditions. In the first, 30% of the population is assumed to have already recovered from the disease, and is therefore immune to it. In the second case, this number is set to 50% [73]. In both cases, we observe the effect of varying vaccination rates (including no vaccination), as well as the times at which schools are reopened.

Fig 7 shows our results for infected children (ages 0–18) and adults (ages 18+) as a function of time. We find that for children, reopening schools could lead to a substantial increase in cases with an initial recovered fraction of 30%. In this case, opening schools earlier leads to a greater increase in cases among children. However, with an initial recovered fraction of 50%, this increase is substantially lower. In all simulations with these initial conditions, reopening schools does not lead to any significant increase in infections among adults. As a consequence,

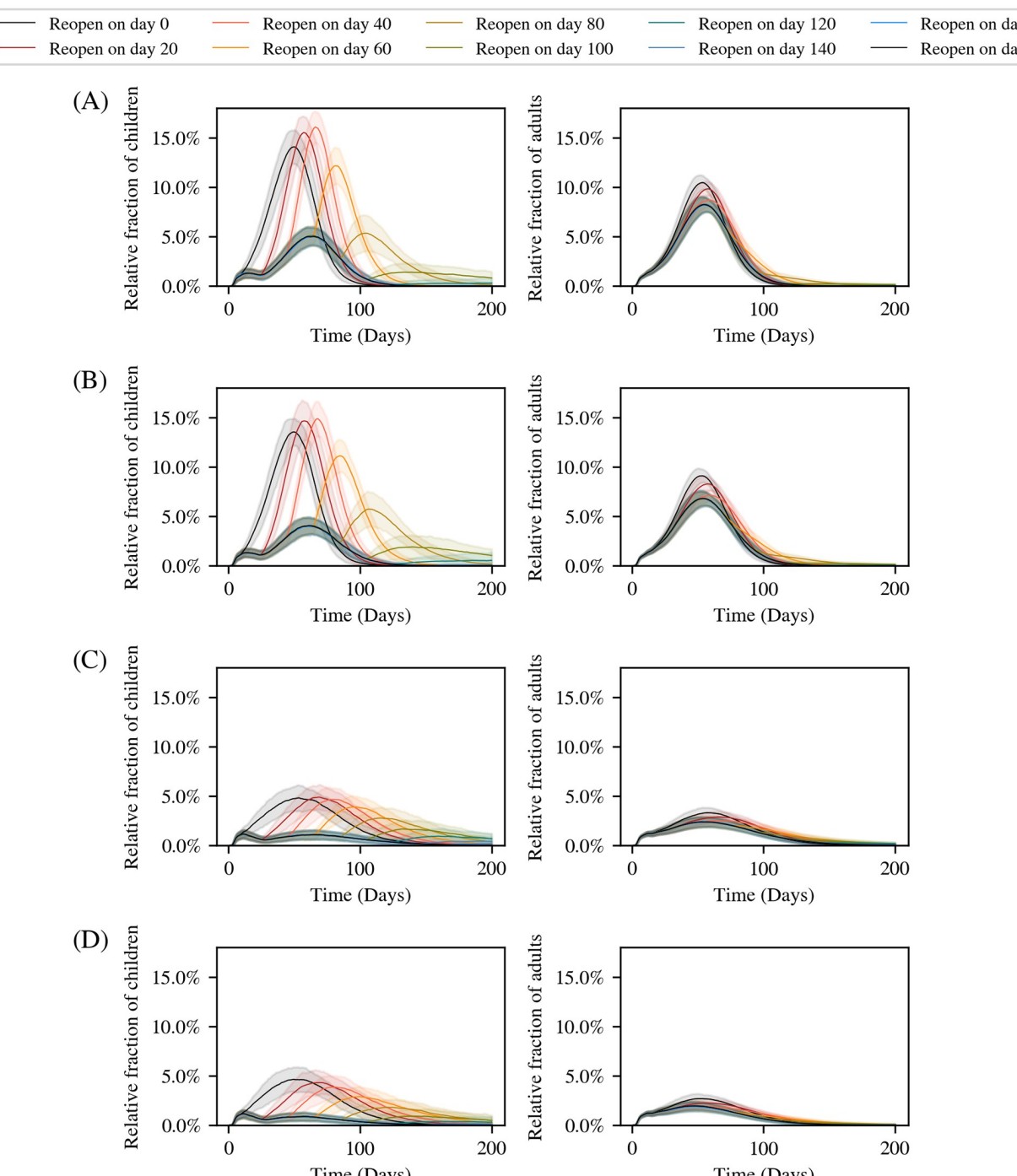

**Fig 7. Simulating different school reopening timelines.** The left-panels show results for children (≤18 years) while the right-panels show the results for adults (>18 years). We begin with an initial recovered fraction of 30% of the population, and with 20% of the population vaccinated (10% with the first dose and 10% with both doses). We compare two different scenarios: in (A) we simulate the population in the absence of a vaccination drive, and in (B) with a daily vaccination rate of 0.4%. The initial plateau in active cases for children is due to the fact that a small fraction of children are infected as part of the initial infection seed, but are not able to infect each other since schools are closed. In (C) and (D) we compare the same scenarios (no vaccination drive and 0.4% daily vaccination rate respectively), but with an initial recovered fraction of 50%. In all cases, opening schools earlier leads to a rise in cases among children, but no significant increase in infections among adults. The curves are all averaged over 100 simulation runs, with the error bars corresponding to 1.96σ.

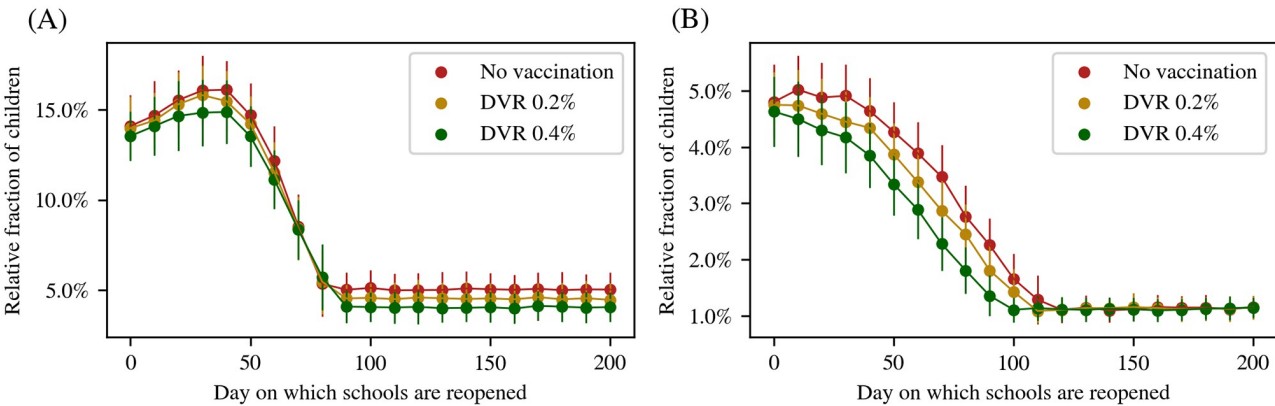

**Fig 8. Effect of schools reopening on the epidemic peak in children.** The peak number of children infected (normalized to their relative population) under different reopening times as indicated by the $x$−axis and different rates of immunization as indicated by the colour of the plots. The two panels show the results (A) assuming 30% initial recovered (B) assuming 50% initial recovered. The values are all averaged over 100 simulation runs, with the error bars corresponding to 1.96$\sigma$.

the total number of severe infections, hospitalizations, and deaths remains largely unaffected when schools are reopened under any of these scenarios.

However, any increase of cases among children, even if asymptomatic or mild, may also be a matter of concern. Thus, we also studied the peak number of cases in children as a function of the time at which schools are reopened. This is shown in Fig 8 for initial recovered fractions of 30% and 50%. From these plots, we see that after a certain time, keeping schools closed makes only a marginal difference to the peak number of infections amongst children. This threshold time depends on both the daily vaccination rate and the initial seropositivity. In all cases, this threshold lies well past the date of peak of the infection that would have occurred if schools had been closed throughout. Thus, if cases in children have already reduced significantly from the peak value, reopening schools would only make a marginal difference.

Another useful criterion for this threshold is obtained by examining the vaccine coverage and net seropositivity as a function of time (see Table 5). Here we define vaccine coverage as the fraction of the total population (i.e., both adults and children) that has received at least one dose. We define net seropositivity as the fraction of the total population that has either been infected and recovered, or has been vaccinated by at least one dose. For this to hold true, the net infection-induced seropositivity among children must be comparable to that among adults.

Comparing Fig 8 with Table 5, we conclude that in order to minimize the peak number of infections in children, schools should be unlocked only when the net seropositivity reaches the range 75–85%. This net seropositivity will be achieved by a combination of natural infections and vaccination. Therefore, vaccinating the population at a higher rate will allow one to open earlier, as will an initially higher seropositivity. Our model allows us to compute these thresholds for any arbitrary vaccination schedule and prior history of infections.

Our results differ in some aspects from those in Ref [74], a modelling study that explores scenarios of partial, progressive, or full school reopening in France, finding somewhat larger severe disease than reported here. We note, however, that the fraction of the population in the 65+ category is over 21% in France, while it is less than 7% in India. Given the strong dependence of the IFR on age, we would expect mortality arising from school reopenings to impact the population far more substantially in France in comparison to India [75]. A different

**Table 5. Vaccine coverage and net seropositivity when schools are reopened.** The fraction of the population that have received at least one vaccine dose (vaccine coverage) and the net seropositivity (both infection and vaccination derived) are tabulated at the moment schools are reopened, for simulations with daily vaccination rates (DVR) of 0% (no vaccination), 0.2% and 0.4%. The top table shows results for an initial recovered fraction of 30% and the bottom corresponds to an initial recovered fraction of 50%.

| DVR | Unlock Schools on Day | 0 | 20 | 40 | 60 | 80 | 100 | 120 | 140 | 160 | 180 | 200 |
|---|---|---|---|---|---|---|---|---|---|---|---|---|
| 0% | Vaccine Coverage | 20.0 | 20.0 | 20.0 | 20.0 | 20.0 | 20.0 | 20.0 | 20.0 | 20.0 | 20.0 | 20.0 |
| | Net Seroprevalence | 44.8 | 48.0 | 58.3 | 71.0 | 77.8 | 79.5 | 79.7 | 79.8 | 79.8 | 79.8 | 79.8 |
| 0.2% | Vaccine Coverage | 20.0 | 23.0 | 26.1 | 29.3 | 32.4 | 35.6 | 38.7 | 41.9 | 45.0 | 48.2 | 51.3 |
| | Net Seroprevalence | 44.8 | 50.0 | 60.7 | 73.1 | 79.6 | 81.8 | 82.9 | 83.7 | 84.5 | 85.1 | 85.8 |
| 0.4% | Vaccine Coverage | 20.0 | 26.1 | 32.4 | 38.8 | 45.2 | 51.6 | 58.0 | 64.3 | 70.7 | 77.1 | 83.5 |
| | Net Seroprevalence | 44.8 | 51.9 | 63.6 | 75.4 | 81.6 | 84.5 | 86.5 | 88.2 | 89.8 | 91.5 | 93.1 |
| 0% | Vaccine Coverage | 20.0 | 20.0 | 20.0 | 20.0 | 20.0 | 20.0 | 20.0 | 20.0 | 20.0 | 20.0 | 20.0 |
| | Net Seroprevalence | 60.8 | 62.7 | 66.2 | 70.0 | 73.1 | 75.1 | 76.1 | 76.5 | 76.8 | 76.9 | 76.9 |
| 0.2% | Vaccine Coverage | 20.0 | 23.0 | 26.1 | 29.3 | 32.4 | 35.6 | 38.7 | 41.9 | 45.0 | 48.2 | 51.3 |
| | Net Seroprevalence | 60.8 | 64.1 | 68.6 | 72.9 | 76.0 | 78.3 | 80.0 | 81.2 | 82.3 | 83.2 | 84.2 |
| 0.4% | Vaccine Coverage | 20.0 | 26.1 | 32.4 | 38.8 | 45.2 | 51.6 | 58.0 | 64.3 | 70.7 | 77.1 | 83.5 |
| | Net Seroprevalence | 60.8 | 65.5 | 71.1 | 76.1 | 79.7 | 82.6 | 85.1 | 87.1 | 89.2 | 91.3 | 93.3 |

modelling study using a compartmental model with India-specific assumptions yields results very similar to those described here [57].

## 3.3 Modeling multiple strains with variable immunity

Finally, we study how the trajectory of a disease is influenced by the introduction of a new and more transmissible strain. The two strains are modelled as identical, except for the parameter $\beta$. We wish to examine how levels of immunity granted by both infection- and vaccination-induced immunity alter the disease trajectory of the second strain.

We consider a community of 10,000 unvaccinated individuals. We initially seed the population with 10 individuals infected by the first strain. The infection is then allowed to propagate without intervention. Once the infections due to this strain have died out, we then introduce the more transmissible second strain by seeding 100 individuals at random.

Individuals who recover from infections with the first strain have their disease trajectories following a second infection altered in two ways: (i) they have a lower relative risk of infection when they come in contact with infected individuals (modeled by introducing a "reinfection probability" parameter), and (ii) prior infection reduces an individual's risk of severe illness, with a protection equal to a single vaccine dose [57].

The trajectory of the second-wave of the disease, induced by the more transmissible variant, is shown in Fig 9A. The worst case occurs when the probability of reinfection is 100%. In this case, since the variant is more transmissible and can infect the entire population, the peak in infections is substantially higher than that of the first-wave. As the reinfection probability is reduced, this peak is diminished.

We can further study the effect that vaccination has on the disease trajectory by delaying the start of the vaccination drive. In Fig 9A, no vaccination drive was included, but we can modify this. Fig 9B shows what happens when the start of the vaccination drive is delayed by successive intervals of 50 days. For simplicity, only the curve for 100% reinfection is shown.

As can be expected, delaying the start of the vaccination drive causes the second, more virulent, strain to spread much faster, and reach a higher peak earlier. The trend is identical for lower daily vaccination rates, although the relative difference between the peaks is less noticeable. Because our model can keep track of each agent's infection history, this study could also

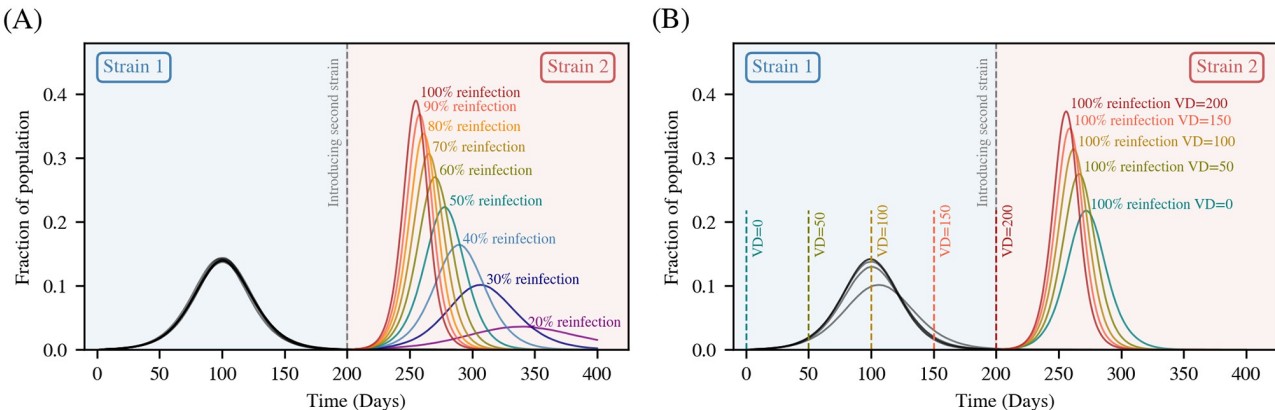

**Fig 9. Effect of reinfection probability on disease progression.** A second more virulent strain is introduced once the first strain has died out. Individuals who contracted the first strain have a reduced relative risk of infection to the second strain, representing a "reinfection probability". In (A) the reinfection probability is varied, producing different disease trajectories. As the reinfection probability reduces, so does the peak of the second wave. In (B) we study the effect of a delayed vaccination drive on the disease trajectory. A constant daily vaccination rate of 0.5% is used. For simplicity, only the curve for 100% reinfection is shown, although the trend is similar for other values of the reinfection probabilities. Starting the vaccination drive later causes the more virulent strain to spread much faster, reaching a much higher peak earlier.

be extended to simulate any complex sequence of variants with given epidemiological characteristics, as discussed in [76–78], for example.

## 4 Discussion

This paper has described the construction and applications of BharatSim, an agent-based simulation framework designed for India. The source code for BharatSim is available from the website www.bharatsim.ashoka.edu.in. The design of BharatSim includes the construction of a detailed synthetic population based on survey and census data for India, a simulation framework that incorporates this population, as well as a tool that can be used to easily visualize the simulations. We applied BharatSim to three specific questions in the context of COVID-19 in India: (i) How do local and global measures (e.g. lockdowns and vaccination drives) affect the disease trajectory of the epidemic in a model Indian city? (ii) When is it feasible to reopen schools during an epidemic? (iii) How does partial immunity arising from a previous infection or vaccination affect the trajectory of a new immune-evasive variant? All of these questions have attracted attention from the point of view of policy.

BharatSim provides an independent way of assessing multiple counterfactual scenarios in which these questions may be addressed. Spatially resolved individual-level models like BharatSim provide a powerful way of describing the spread of an infectious disease but are not restricted to this context. They are particularly useful for discussions of policy, because insights into social and individual behaviour can be incorporated into them. The complexity of the underlying dynamical contact networks is also easily incorporated into models of this type.

We found that the time of initiation and the vaccination rate associated with a vaccination drive can be tuned so as to have the maximum effect on the disease trajectory. Additionally, we established that at the levels of seropositivity reported across much of India in the wake of the first and second waves of COVID-19, schools could have been reopened with relatively little impact several months before they were actually opened. Finally, we found that the effects of a vaccination drive, if timed appropriately, could blunt the increase of cases due to a second, more transmissible, viral strain.

BharatSim, although applied only to COVID-19 in the scenarios described here, lends itself to more versatile use. Describing the direct transmission of any other disease is straightforward. We have also used BharatSim recently to study how seasonal variations in contact patterns affect the transmission of an influenza-type disease, using contact matrices obtained from real-world data. This work has appeared in Ref [81] and is part of an ongoing investigation into the seasonality of influenza in northern India.

Adding components describing environmental variables, BharatSim can be extended to describe, say, the transmission and spread of typhoid within communities. This would require a parametrization both of hydrology and water-use practices, tied to geographical variables. Adding in the life cycle and dynamics of arthropod vectors such as mosquitoes to the dynamics of infected agents, allows the modelling of vector-borne diseases such as malaria and dengue [82–85]. Mosquitoes could also be modelled as agents [86, 87], but a more practical description might be in terms of fields which interact dynamically with human agents [86, 88]. Indeed, beginning with assumptions about individual behaviour as well as environmental and social determinants, one could potentially describe the incidence of non-communicable diseases in time for a prescribed population, calibrating these results against available data.

To demonstrate the versatility of BharatSim beyond the applications described here, we have further used it to model the spread of mpox in a population of men who have sex with men (MSMs) and their associated sexual and non-sexual networks. To our knowledge, only two agent-based models of mpox are present in the literature [79, 80]). Our results, described in S9 Appendix are the following: if contact is solely via MSM networks, we see a fast spread across this network that rises and falls over a relatively fast time-scale. If, in addition, we assign a low probability of transmission between agents in the household of infected MSMs, we see a slower rise and fall in cases that also peaks at a later time. The nature of the spread depends upon the level of the initial infection in the non-MSM population. The interaction of the MSM and non-MSM population can lead to an unusual two-peak structure in the numbers of infected with time when the infection is initially dominated by non-MSM agents. Expanding the possibility of such transmission to work networks can lead to a situation of prolonged prevalence.

Agent-based models are powerful because the behaviour of large groups of individual actors can be emergent as a consequence of their interactions. The flexibility of such models has led to their use in a number of fields [89]. They have been used to study large-scale economic networks and financial systems [90]. In the social sciences, agent-based models have been developed to study opinion dynamics, inequality, cooperation, and social norms [91, 92]. They have also been used to model civil and domestic violence [93, 94]. Such computational approaches are especially interesting because they expand the capabilities of standard economic and game-theoretic modelling [95]. Because its synthetic population is specific to India, BharatSim should find novel uses in the social sciences in this context. To the best of our knowledge, these have not been previously explored.

The methodology of agent-based models is particularly useful in problems when agents can be programmed to choose from a range of behaviours in response to exogenous information. These may derive from their interactions with a limited number of other agents or from environmental variables. The tendency of sub-groups of individuals within a society to interact largely within themselves, for example groups of individuals bound together through social or economic ties or caste affiliations, can be described in terms of the interaction networks that define the synthetic population. Thus, frameworks such as BharatSim provide us with a way of potentially assessing the social determinants of disease, through information, usable within models, about how individuals make decisions while embedded in their social and economic contexts. We note too that while BharatSim comes with a synthetic population that is India-

specific, nothing prevents the use of the simulation and visualization engines on a similarly constructed population for any part of the world.

A particularly powerful use of BharatSim might be the following: India has initiated the single largest digital public-health initiative in the world, the Ayushman Bharat Digital Mission (ABDM) [96]. Its broad aim is the development of an integrated digital health infrastructure. The ABDM will create a national digital health ecosystem that supports Universal Health Coverage. Data, information and services integrated under the ABDM will be accessible via an open, interoperable digital system. Every Indian will have a digital health ID, the ABHA number. A comprehensive repository of all health facilities in India across different systems of medicine, including public and private health facilities, will be completed in parallel.

The debate around the ABDM has centred around two main questions: of privacy and, separately, of the digital divide. Concerns with privacy associated with the direct use of ABDM data could potentially be alleviated if these data were to be made the basis for high-quality synthetic populations. Once such a population is constructed and rigorously benchmarked, running scenarios that can be tweaked in real-time becomes possible. Regarding the digital divide, one could hope to use these methods to identify populations that are under-served, potentially correlating these with social determinants of health that would need to be addressed in parallel.

These possibilities for further investigation point to the usefulness of methods such as BharatSim to interrogate specific policy interventions in health and in those areas that overlap it. These could draw from what we know to be the social and economic determinants of health in India and similar LMICs. To our knowledge, such areas of investigation remain relatively unexplored, particularly in the complex social environment that India presents.

## Supporting information

**S1 Appendix. Scaling BharatSim with population size and model complexity.** The simulation times for different population sizes are compared, both for a simple SIR model and the more complicated INDSCI-SIM model described here. We show that the BharatSim simulation engine can handle populations of close to 10 million agents, and that the simulation time scales up as $\mathcal{O}(N)$.
(PDF)

**S2 Appendix. Creating and benchmarking a synthetic population for Mumbai.** We benchmark the quality of the synthetic population by comparing the statistics of the numerical and categorical data in both the synthetic population and survey data. In each case, a similarity measure is reported.
(PDF)

**S3 Appendix. Description of the compartmental model.** The disease-progression in a single well-mixed compartmental model is briefly described, and an association is made between the results of this compartmental model and a well-mixed (all-to-all connected) agent-based model with the same compartments.
(PDF)

**S4 Appendix. Sensitivity analyses.** We study in detail how sensitive our results are to our model choices. We find that our main results are quite robust to the choice of agents' travel distances, workplace occupancies, and time spent at home.
(PDF)

**S5 Appendix. The effect of vaccination.** We show how vaccination affects both the relative risks of infection, and of contracting severe disease. These risks are governed by modifying individual agents' $\beta$ and $\alpha$ parameters respectively. When an agent receives a dose of the vaccine, their protection increases linearly over a 14 day window, until it plateaus to the maximum protection offered by that dose. The reduction in the relative risk of infection (i.e. in $\beta$) is assumed to be the same for each age-group, while the reduction in the risk of contracting severe disease is age-stratified.
(PDF)

**S6 Appendix. Vaccine dose-prioritization and avoiding wastage.** The vaccine doses are distributed between first- and second-shot vaccinations in an 80:20 ratio, in such a way as to minimize dose-wastage, as discussed in the main text. We discuss the algorithm used to enforce this, and illustrate through examples how the doses are distributed in a population of 10,000 agents for different daily vaccination rates.
(PDF)

**S7 Appendix. Describing the populations used in our simulations.** We describe two of the populations used to obtain our simulation results. The first is a population for the entire city of Pune with 3.13 million individuals. The second is a section of the synthetic population for the city of Pune, which is chosen with 20,316 individuals with 6500 homes, 120 workplaces, and 1 school.
(PDF)

**S8 Appendix. Using BharatSim for epidemic forecasts: A case study for Pune.** We compare our model projections with real data for confirmed cases from the initial part of the first wave of COVID-19 in the city of Pune. These projections, and an understanding of how they are modulated by the interventions made, can then be directly used for public health planning in an epidemic situation.
(PDF)

**S9 Appendix. Studying the spread of mpox.** We use BharatSim to model the spread of mpox, a viral infection caused by a zoonotic virus in the genus *Orthopoxvirus*. A global outbreak of mpox associated with sexual contact has been ongoing since May 2022, with the vast majority of cases being diagnosed among men who have sex with men (MSMs). Here, we describe a possible use of BharatSim to study the spread of mpox in a population of 10,000 individuals that is meant to represent a group of MSMs and their contact networks.
(PDF)

## Acknowledgments

GIM and PC are grateful to Brian Wahl, Sandeep Krishna, Farhina Mozaffer, Sandeep Juneja, Harish Iyer, Kayla Laserson, Anand Krishnan, Rajesh Sundaresan, Siva Athreya, Mihir Arjunwadkar, and Joy Monteiro for many discussions. GIM acknowledges the Tata Institute of Fundamental Research for an Adjunct Professorship. The authors acknowledge the use of computational facilities provided by the Centre for Bioinformatics and Computational Biology at Ashoka University. Yerik Singh contributed to the online documentation, in particular the section on the Simulation Engine, and we thank him for his help. The Thoughtworks team would like to acknowledge the sponsors of the Engineering for Research initiative for supporting the development of the *EpiRust* simulator.

## Author Contributions

**Conceptualization:** Philip Cherian, Harshal Hayatnagarkar, Chhaya Yadav, Debayan Gupta, Gautam I. Menon.

**Data curation:** Jayanta Kshirsagar, Bhavesh Neekhra, Harshal Hayatnagarkar, Kshitij Kapoor, Chandrakant Kaski, Ganesh Kathar, Swapnil Khandekar, Saurabh Mookherjee, Praveen Ninawe, Pranjal Ranka, Vaibhhav Sinha, Tina Vinod, Debayan Gupta.

**Formal analysis:** Philip Cherian, Bhavesh Neekhra, Harshal Hayatnagarkar, Saurabh Mookherjee, Praveen Ninawe, Riz Fernando Noronha, Pranjal Ranka, Vaibhhav Sinha, Tina Vinod, Debayan Gupta, Gautam I. Menon.

**Funding acquisition:** Debayan Gupta, Gautam I. Menon.

**Investigation:** Philip Cherian, Jayanta Kshirsagar, Bhavesh Neekhra, Gaurav Deshkar, Harshal Hayatnagarkar, Kshitij Kapoor, Chandrakant Kaski, Ganesh Kathar, Swapnil Khandekar, Saurabh Mookherjee, Praveen Ninawe, Riz Fernando Noronha, Pranjal Ranka, Vaibhhav Sinha, Tina Vinod, Debayan Gupta, Gautam I. Menon.

**Methodology:** Philip Cherian, Jayanta Kshirsagar, Bhavesh Neekhra, Gaurav Deshkar, Harshal Hayatnagarkar, Chandrakant Kaski, Ganesh Kathar, Swapnil Khandekar, Saurabh Mookherjee, Praveen Ninawe, Riz Fernando Noronha, Pranjal Ranka, Vaibhhav Sinha, Tina Vinod, Debayan Gupta, Gautam I. Menon.

**Project administration:** Harshal Hayatnagarkar, Praveen Ninawe, Chhaya Yadav, Debayan Gupta, Gautam I. Menon.

**Resources:** Philip Cherian, Jayanta Kshirsagar, Bhavesh Neekhra, Gaurav Deshkar, Harshal Hayatnagarkar, Chandrakant Kaski, Ganesh Kathar, Swapnil Khandekar, Saurabh Mookherjee, Praveen Ninawe, Riz Fernando Noronha, Pranjal Ranka, Vaibhhav Sinha, Chhaya Yadav, Debayan Gupta, Gautam I. Menon.

**Software:** Philip Cherian, Jayanta Kshirsagar, Bhavesh Neekhra, Gaurav Deshkar, Harshal Hayatnagarkar, Kshitij Kapoor, Chandrakant Kaski, Ganesh Kathar, Swapnil Khandekar, Saurabh Mookherjee, Praveen Ninawe, Riz Fernando Noronha, Pranjal Ranka, Vaibhhav Sinha, Tina Vinod, Chhaya Yadav, Debayan Gupta.

**Supervision:** Philip Cherian, Jayanta Kshirsagar, Harshal Hayatnagarkar, Praveen Ninawe, Chhaya Yadav, Debayan Gupta, Gautam I. Menon.

**Validation:** Philip Cherian, Jayanta Kshirsagar, Bhavesh Neekhra, Gaurav Deshkar, Harshal Hayatnagarkar, Kshitij Kapoor, Chandrakant Kaski, Ganesh Kathar, Swapnil Khandekar, Saurabh Mookherjee, Praveen Ninawe, Riz Fernando Noronha, Pranjal Ranka, Vaibhhav Sinha, Tina Vinod, Debayan Gupta, Gautam I. Menon.

**Visualization:** Philip Cherian, Jayanta Kshirsagar, Bhavesh Neekhra, Harshal Hayatnagarkar, Kshitij Kapoor, Chandrakant Kaski, Ganesh Kathar, Swapnil Khandekar, Saurabh Mookherjee, Praveen Ninawe, Riz Fernando Noronha, Pranjal Ranka, Vaibhhav Sinha, Tina Vinod, Debayan Gupta, Gautam I. Menon.

**Writing – original draft:** Philip Cherian, Jayanta Kshirsagar, Bhavesh Neekhra, Harshal Hayatnagarkar, Debayan Gupta, Gautam I. Menon.

**Writing – review & editing:** Philip Cherian, Gaurav Deshkar, Harshal Hayatnagarkar, Kshitij Kapoor, Chandrakant Kaski, Ganesh Kathar, Swapnil Khandekar, Saurabh Mookherjee,

Praveen Ninawe, Riz Fernando Noronha, Pranjal Ranka, Vaibhhav Sinha, Tina Vinod, Chhaya Yadav, Debayan Gupta, Gautam I. Menon.

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
