## [Decision Letter · Decision Letter 0]

18 Dec 2023

Dear Prof. Menon,

Thank you very much for submitting your manuscript "BharatSim: An agent-based modelling framework for India" for consideration at PLOS Computational Biology.

As with all papers reviewed by the journal, your manuscript was reviewed by members of the editorial board and by several independent reviewers. In light of the reviews (below this email), we would like to invite the resubmission of a significantly-revised version that takes into account the reviewers' comments.

We cannot make any decision about publication until we have seen the revised manuscript and your response to the reviewers' comments. Your revised manuscript is also likely to be sent to reviewers for further evaluation.

Sincerely,

Samuel V. Scarpino

Academic Editor

PLOS Computational Biology

Virginia Pitzer

Section Editor

PLOS Computational Biology

Reviewer's Responses to Questions

**Comments to the Authors:**

Reviewer #1: I thank the authors for the opportunity to read their informative manuscript “BharatSim: An agent-based modelling framework for India”. In this work, Cherian et al. develop a novel agent-based modeling (ABM) framework BharatSim. Crucially, this ABM contains pre-specified data and defaults for India, allowing for easier and more accurate scenario projections for this unique context. The ABM is also quite efficient and has extensive documentation and support. The scenarios developed and discussed in the manuscript include the impact of various NPIs and timing of school re-opening. These scenarios are relevant to development of public health planning and guidance; modern open-source computational tooling is quite a valuable step.

I found the software development described in this manuscript quite impressive and appreciated the thoughtful documentation. I can see substantial research value that will be derived from this work. In addition, the harmonization of the disparate Human Development Survey, Census, GPW population density, and others is quite valuable. Both are important contributions.

However, I have a number of methodological concerns that should be addressed. These include insufficient sensitivity analyses, unclear modeling assumptions, and conclusions that do not match the strength of the evidence. I have split my main concerns into a major and minor section below:

Major

- It is my understanding that many of the datasets used in the manuscript are quite out of date and may not reflect the current Indian population. The clearest example of this may be the Census data, which I believe are from 2011 and likely do not accurately reflect India’s fast-growing population to the resolution used in this manuscript. There are similar issues with the Gridded Population of the World, the Human Development Survey, and the IHDS-II datasets. Some form of statistical correction and/or additional supplementary analyses examining the sensitivity of results to out-of-date data would be justified. I believe this is a crucial issue because one of the main advances in this manuscript is combining these datasets and producing results uniquely appropriate for the Indian context.

- I believe substantially more sensitivity analyses are needed. There are strong assumptions on the population distributions (uniform inside the population grids I believe?), the number of movements and distance moved between “home” and “work” patches, and the specification of the epidemiological scenarios. While I appreciate that these models are computationally intensive to run, it is impossible to evaluate the modeling framework without understanding which assumptions are robust and which change the results. There are certainly parameters to which the results are not robust — I believe those should be examined and reported.

- In all the epidemic scenarios, I am uncomfortable with the framing of the results. These results are not statistical fits to epidemic data and do not necessarily correspond to observed epidemic dynamics. These scenario-based models can project dynamics subject to particular assumptions. Greater emphasis on and transparency of the necessary assumptions would make these results stronger and more believable. For example, I am unclear what key assumptions drive the finding in the school reopening example. It seems quite different than results in other settings (see e.g., https://doi.org/10.1038/s41467-021-21249-6). These deviations are not explained or contextualized and, to me, make the results less believable.

- I find the number of figures overwhelming. It would help if some of the model explanation and figures could move to the appendix to streamline the paper. I had trouble following which pieces were important for the model and which were ancillary choices.

Minor

- I don’t understand the framing in the Introduction of compartmental models vs. network-based models or ABMs (see: “Both network- and agent-based models must use many more assumptions than compartmental models require.”). As I understand it, the model used in BharatSim is an extension of the SIR compartment model which generalizes the mixing of individuals within the population. This is a classic framing, but one I would still describe as compartment-based.

- I do not agree with the framing around “realism”, well-mixed populations, and ODE approaches. These ABMs allow for extraordinarily detailed scenarios and investigation of complex dynamics, but they are fundamentally scenario projections. There is no fit to data and the results are not necessarily realistic. I don’t see that as a flaw — on the contrary, it can be a real strength and allow us to examine unobserved possibilities.

- Broader engagement with the literature for each of the scenarios would be appropriate. For example, there is a broad literature in variant dynamics (see e.g., 10.1126/science.abj0113, https://doi.org/10.1098/rsif.2021.0900, DOI: 10.1126/science.abl9551) and it would be useful to understand how the approaches used here fit in with this broader literature — with similar efforts for the other scenarios.

- I would appreciate if the supplement could include definitions and model values associated with each parameter in the supplement "S3: Description of the compartmental model"

- I would appreciate additional supplementary figures and analysis of the joint distribution between population parameters. Which parameters are correlated? Is it the case that the parameters we expect to be uncorrelated are actually uncorrelated?

Overall, I found the modeling software and data harmonization described in this work to be a valuable contribution. I appreciated the excellent software development and thoughtful model development described in the manuscript.

Reviewer #2: 1) In the introduction a proper survey of agent-based models used for studying other infectious diseases should be included.

2) In Section 2.1.2, how are capturing simple attributes like height, weight, age and sex relevant to the study of Covid-19. This should be supported using references. Why not include attributes like blood group, nutrition etc. which may be more relevant.

3) There is no validation of the contact patterns, workplaces, travel using source/survey data. Only validation for simple attributes have been included. Authors should also validate the other assumptions in the synthetic populations.

4) What are these numbers in Table 2? Are these correlations?

5) The evaluation of the model is based only on simulations. Why not use real data to verify the predicted infection trajectories wrt the interventions? The authors should include a comparison based on real data.

6) On page 26, the authors claim that the model can be applied to other infectious diseases like typhoid. I would suggest that the authors include such an example based on typhoid. This would make the manuscript more applicable as studying Covid-19 at this point does not have much applicability.

Reviewer #3: Review uploaded as an attachment.

**Have the authors made all data and (if applicable) computational code underlying the findings in their manuscript fully available?**

Reviewer #1: Yes

Reviewer #2: Yes

Reviewer #3: None

PLOS authors have the option to publish the peer review history of their article (what does this mean?). If published, this will include your full peer review and any attached files.

Reviewer #1: No

Reviewer #2: No

Reviewer #3: No
---

## [Decision Letter · Decision Letter 1]

15 Jul 2024

Dear Prof. Menon,

Thank you very much for submitting your manuscript "BharatSim: An agent-based modelling framework for India" for consideration at PLOS Computational Biology.

As with all papers reviewed by the journal, your manuscript was reviewed by members of the editorial board and by several independent reviewers. In light of the reviews (below this email), we would like to invite the resubmission of a significantly-revised version that takes into account the reviewers' comments.

I agree with both reviewers that this work shows quite a bit of promise, but there is still a need for substantial revision to just the claims made in the paper. The authors would either need to significantly narrow the scope of their work or include a more detailed analysis using the kinds of data described by reviewer 4 (who was a previous reviewer) and also extend the analysis to other pathogens (as discussed by reviewer 2). My strong sense is that the later (the use of more detailed data and extension to other pathogens) is what's really needed here. I want to stress the importance of addressing this concerns in a revision.

We cannot make any decision about publication until we have seen the revised manuscript and your response to the reviewers' comments. Your revised manuscript is also likely to be sent to reviewers for further evaluation.

Sincerely,

Samuel V. Scarpino

Academic Editor

PLOS Computational Biology

Virginia Pitzer

Section Editor

PLOS Computational Biology

I agree with both reviewers that this work shows quite a bit of promise, but there is still a need for substantial revision to just the claims made in the paper. The authors would either need to significantly narrow the scope of their work or include a more detailed analysis using the kinds of data described by reviewer 4 (who was a previous reviewer) and also extend the analysis to other pathogens (as discussed by reviewer 2). My strong sense is that the later (the use of more detailed data and extension to other pathogens) is what's really needed here. I want to stress the importance of addressing this concerns in a revision.

Reviewer's Responses to Questions

**Comments to the Authors:**

Reviewer #2: 1) The evaluation of the model is based only on simulations. Why not use real data to verify the predicted infection trajectories wrt the interventions? The authors should include a comparison based on real data.

This comment has not been addressed. The authors need to give an assessment of their results with some real world data.

2)The model should be extended to include one other infectious disease to show applicability to modelling infectious diseases other than covid.

This comment was made earlier also but not addressed.

Reviewer #4: Cherian et al. present a new software package BharatSim, which is an agent-based modeling framework tailored to India. This ABM is applied to a number of different scenarios, demonstrating its versatile parameterization and the wide range of provided data sources.

Despite the impressive software engineering described in the manuscript, I remain uncomfortable with the presentation of the results and unconvinced by the underlying dataset. Although the additional sensitivity analyses are helpful, I do not believe the ameliorate the concerns around the accuracy of the data for specific policy recommendations. I elaborate on my two main concerns below:

• The modeling framework is geared to answer policy questions and describe dynamics in a named Indian city. The findings are used to make specific policy recommendations and quite generic usage is advocated in the Discussion. I am not convinced that this model can be used to answer these specific, detailed policy questions without specific, detailed data and numerical fitting to that data. Rather, this model can be used to examine generic questions subject to specific modeling assumptions and the results would not necessarily be applicable to any particular real-world context. I believe the results and particularly the Discussion need substantial additional edits to change the level of specificity in the framing and recommendations.

• The data are not specific and detailed to the current context. In general, older datasets are useful and provide value, but not when answering specific, detailed modeling questions on recent disease dynamics. I agree that it is appropriate to use these datasets as the best available data, but I still do not understand which modeling assumptions are influential. This question was not answered by the additional sensitivity analyses. In particular, I have trouble figuring out what changes in an updated dataset would have in this modeling framework. Is it anything? Why or why not? How do these uncertainties change the certainty around policy recommendations?

Although the software engineering and data collation described in this manuscript are valuable, I remain unconvinced by the presentation of results. This concern is further exacerbated by the age of the underlying data for the specific policy recommendations presented.

**Have the authors made all data and (if applicable) computational code underlying the findings in their manuscript fully available?**

Reviewer #2: None

Reviewer #4: Yes

PLOS authors have the option to publish the peer review history of their article (what does this mean?). If published, this will include your full peer review and any attached files.

Reviewer #2: No

Reviewer #4: No
---

## [Decision Letter · Decision Letter 2]

2 Dec 2024

Dear Prof. Menon,

We are pleased to inform you that your manuscript 'BharatSim: An agent-based modelling framework for India' has been provisionally accepted for publication in PLOS Computational Biology.

Best regards,

Samuel V. Scarpino

Academic Editor

PLOS Computational Biology

Virginia Pitzer

Section Editor

PLOS Computational Biology

Feilim Mac Gabhann

Editor-in-Chief

PLOS Computational Biology

Jason Papin

Editor-in-Chief

PLOS Computational Biology

Reviewer's Responses to Questions

**Comments to the Authors:**

Reviewer #2: All my questions have been answered. The section included for the first wave Pune data and Mpox shows the usefulness of the proposed approach.

**Have the authors made all data and (if applicable) computational code underlying the findings in their manuscript fully available?**

Reviewer #2: Yes

PLOS authors have the option to publish the peer review history of their article (what does this mean?). If published, this will include your full peer review and any attached files.

Reviewer #2: No

---

## [Editor Report · Acceptance letter]

20 Dec 2024

PCOMPBIOL-D-23-01579R2 

BharatSim: An agent-based modelling framework for India

Dear Dr Menon,

I am pleased to inform you that your manuscript has been formally accepted for publication in PLOS Computational Biology. Your manuscript is now with our production department and you will be notified of the publication date in due course.

With kind regards,

Lilla Horvath
